# In Silico Genomic and Metabolic Atlas of *Limosilactobacillus reuteri* DSM 20016: An Insight into Human Health

**DOI:** 10.3390/microorganisms10071341

**Published:** 2022-07-02

**Authors:** Paisleigh Smythe, Georgios Efthimiou

**Affiliations:** 1Centre for Atherothrombosis and Metabolic Disease, Hull York Medical School, Castle Hill Hospital, Daisy Building, Hull HU16 5JQ, UK; p.j.smythe-2018@hull.ac.uk; 2Department of Biomedical and Forensic Sciences, University of Hull, Cottingham Road, Hardy Building, Hull HU6 7RX, UK

**Keywords:** probiotic, health benefits, adhesion, genome, metabolome, reuterin, competition, lactic acid bacteria, *Limosilactobacillus reuteri*

## Abstract

Probiotics are bacterial strains that are known to provide host health benefits. *Limosilactobacillus reuteri* is a well-documented lactic acid bacterium that has been cultured from numerous human sites. The strain investigated was *L. reuteri* DSM 20016, which has been found to produce useful metabolites. The strain was explored using genomic and proteomic tools, manual searches, and databases, including KEGG, STRING, BLAST Sequence Similarity Search, and UniProt. This study located over 200 key genes that were involved in human health benefit pathways. *L. reuteri* DSM 20016 has metabolic pathways to produce acetate, propionate, and lactate, and there is evidence of a pathway for butanoate production through a FASII mechanism. The bacterium produces histamine through the hdc operon, which may be able to suppress proinflammatory TNF, and the bacterium also has the ability to synthesize folate and riboflavin, although whether they are secreted is yet to be explored. The strain can bind to human Caco2 cells through srtA, mapA/cnb, msrB, and fbpA and can compete against enteric bacteria using reuterin, which is an antimicrobial that induces oxidative stress. The atlas could be used for designing metabolic engineering approaches to improve beneficial metabolite biosynthesis and better probiotic-based cures.

## 1. Introduction

Probiotics are bacteria or yeast that can withstand acid and bile, colonizing, surviving, and metabolizing within the human gut to provide health benefits to the host, such as providing digestive comfort [1]. Probiotics can be used to treat chronic diseases, aiding in the modulation of the immune response, and regulating the inflammatory response [2]. Lactic acid bacteria are most commonly used as probiotics, as they are able to ferment, metabolize carbohydrates, and withstand the harsh environment. An analysis of 22 systematic reviews highlighted beneficial probiotic effects through preventing upper respiratory tract infections, diarrheal diseases, necrotizing enterocolitis (NEC), and others [3]. Lactic acid bacteria could be used in the treatment and prevention of colorectal cancers, in topical cosmetics for the treatment of atopic skin disorders, and, in certain cases, offer anti-depressive effects [4,5,6].

Microbial infections are a known cause of cancer, which is a multistep pathology requiring multiple mutations [7]. Microorganisms can damage host DNA, generating instability of the genome and contributing to multistep tumorigenesis. Microbes may also be able to promote tumorigenesis through modulating cancer hallmarks [8]. Probiotics may be able to prevent and treat cancer, due to the beneficial effect they can have on the human host [9]. *L. reuteri* has previously been documented to prevent colorectal cancer in animal models [10], with *L. reuteri* DSM 20016 specifically, although isolated from pig kidneys, able to significantly improve the survival of mice with melanoma tumors [11].

Gastrointestinal symptoms account for 10% of all consultations with general practioners and 14% of the National Health Service drug budget [12]. Gastrointestinal problems could be caused by gut dysbiosis—an imbalance of gut microorganisms. Dysbiosis of the microbiota can lead to an impact on human health, with the intestinal immune system and conditions, such as gluten intolerance, being influenced by certain microbiota abundance [13]. Immune-mediated inflammatory diseases, such as inflammatory bowel disease (IBD), can also be affected with studies having found that those with IBD have 25% fewer microbial genes in comparison to the gut of healthy individuals [14]. Those with subacute intestinal symptoms may also be more likely to have psychiatric disorders, such as anxiety, depression, and panic disorders [15]. *Limosilactobacillus reuteri* is a heterofermentative Gram-positive *Limosilactobacillus* and a potential new probiotic. *L. reuteri* can prevent diarrhea and can help to regulate the immune function of the intestine [16]. It may also function by inhibiting enteric infection severity and modulating lipopolysaccharide-induced inflammation [17]. *L. reuteri* can produce several different biological compounds that could provide benefits to the host, including an antimicrobial molecule known as reuterin and different short chain fatty acids (SCFAs), which can help to modulate the immune system. *L. reuteri* is already being used as a probiotic, with a study noting that the lactic acid bacterium reduced sepsis frequency and hospital stay durations in newborns with infant necrotizing enterocolitis [18]. Whilst another study detailed its use as a therapeutic agent in treating children with acute rotavirus diarrhea [19].

Omics-based research has made many recent advancements, due to the increased availability of whole genome sequences [20]. Genomic and metabolomic investigations into probiotics is a viable strategy, providing a cost-effective and preliminary search into a potential probiotic’s metabolic functions and to explore its mode of actions in relation to human host health benefits. Previous omics investigations have been used to characterize the resistances of *Lactobacillus* and *Bifidobacterium* [21], conduct genomic analyses on other *L. reuteri* strains to identify stress adaptations and antimicrobial profiles [22,23], and determine the safety of *L. reuteri* strains to determine their suitability as probiotics [24]. Figure 1 details the likely health benefits found within the strain.

*L. reuteri* DSM 20016 is a potential candidate as a probiotic. Found within the human gut and in human feces, *L. reuteri* could be well suited for gut colonization. The aim of this report was to create a detailed atlas of the metabolic pathways leading to the formation of beneficial metabolites in this strain. The objectives were to use both genomic and proteomic bioinformatic tools to identify and localize the genes that are involved in these pathways and their encoded proteins. With more research, *L. reuteri* DSM 20016 could be utilizable for treating several inflammatory diseases and other pathologies. Based on this atlas, the personalized metabolic engineering of this probiotic based on the patient’s microbiota profile to improve the biosynthesis of beneficial products could be one area for the utilization of probiotic-based cures for the treatment of human diseases [25].

## 2. Materials and Methods

### 2.1. Bacterial Strain

The whole sequence of the *L. reuteri* DSM 20016 genome has been characterized in GenBank and can be found at ascension number CP000705.1 [26]. The assembly was downloaded at https://www.ncbi.nlm.nih.gov/assembly/GCA_000016825.1 (accessed on 23 June 2020) using the source database of GenBank and the file type of the Genomic GenBank format (.gbff).

### 2.2. Search Strategies and Gene Selection

To identify potential health benefits within *Limosilactobacillus* species, and specifically *L. reuteri* itself, PubMed databases were searched for peer-reviewed articles with no date restrictions. The search terms used were ‘*Lactobacillus*’, ‘*Limosilactobacillus*’, ‘*Lactobacillus reuteri*’, ‘*Limosilactobacillus* reuteri’, ‘*L. reuteri* DSM 20016’, ‘probiotic’, ‘health’, ‘adhesion’, ‘colonisation’, ‘immunomodulation’, ‘biofilm’, and ‘inflammation’. We searched the full texts of these studies to determine areas of health benefits in *Limosilactobacillus* and probiotic-related strains. The cited references of Mu et al., 2018 [27], and Sriramulu et al., 2008 [28], were also searched, due to their investigations into key areas of *L. reuteri* DSM 20016 specifically. Areas of focus were the production of metabolites that had known health benefits in human hosts, such as reuterin, histamine, and SCFAs.

The genome of *L. reuteri* DSM 20016 was searched in the Kyoto Encyclopedia of Genes and Genomes (KEGG; release 91 and 99; https://www.genome.jp/kegg/ (accessed on 2 June 2020 and 8 August 2021)) [29], and the key health-related pathways identified through the meta-analysis were identified in this strain. Complete pathways were synthesized where possible. Unknown pathways or pathways with missing genes were analyzed using bioinformatics.

### 2.3. Bioinformatic Analysis and Gene Annotation

To identify missing genes from key pathways, Basic Local Alignment Search Tool (BLAST; versions 2.10.1, 2.11.0 and 2.12.0) [30] was used to compare the nucleotide sequences of genes of similar strains to that of the *L. reuteri* DSM 20016 sequence and genes. The sequences of these genes were gathered from UniProt (https://www.uniprot.org/ (accessed on 20 June 2020 and 5 September 2021); versions release—6 January 2020 and release-2—4 January 2021) [31] by searching the gene name and finding the gene fasta sequence from the closest related strain. A manual examination of hits facilitated the identification, with parameters set at 30% identity and 50% coverage. In certain cases, the searching of the gene against the genome itself was accomplished using Artemis (release 18.1.0; https://www.sanger.ac.uk/tool/artemis/ accessed on 20 June 2020) [32] and the downloaded assembly sequence.

STRING (Search Tool for the Retrieval of Interacting Genes/Proteins; versions 11 and 11.5; https://string-db.org/ (accessed on 10 September 2020 and 23 April 2022)) [33] was utilized to ensure that there was a strong connection between genes within pathways and to highlight any highly related missing genes. The minimum required interaction score was set to 0.400 (medium confidence) and 0.700 (high confidence), and the maximum number of interactors to show was set to 10. 

Collated key genes were organized into their respective health benefit pathways in Appendix A, including previously uncharacterized genes. Finally, key genes were plotted on the genome using DNAPlotter (release 18.1.0; https://www.sanger.ac.uk/tool/dnaplotter/ (accessed on 28 June 2020)) [34] for visualization using the assembly sequence previously characterized.

## 3. Results and Discussion

*Limosilactobacillus reuteri* DSM 20016 contains only one chromosome that is 1,999,618 nucleotides in size, encompassing 1900 genes. There is a GC content of 38.9%. Figure 2 details a genomic map of *L. reuteri* DSM 20016.

*L. reuteri* DSM 20016 has a genomic sequence very similar to that of *L. reuteri* JCM 1112, which is a sequence also derived from the original F275, before laboratory cultivation. *L. reuteri* JCM 1112 has two unique regions between 442995–451429 (LAR_0380–LAR_0385) and 1064161–1094397 (LAR_0936 to LAR_0958). The first unique region encodes enzymes involved in glycolysis. The second unique region encodes nitrate reductase subunits and molybdopterin biosynthesis genes.

The first unique region of *L. reuteri* JCM 1112 has homologs in *L. reuteri* DSM 20016 apart from glyceraldehyde-3-P dehydrogenase, which cannot be located. These strains are commonly confused, suggesting that the resequencing of strains before studies should be strongly encouraged. The implications of the missing nitrate genes have not been explored, but it would suggest that, while *L. reuteri* JCM 1112 is capable of metabolizing extracellular and intracellular nitrate, *L. reuteri* DSM 20016 is not.

There are variable sources as to the number of protein-coding genes within these strains, with KEGG utilizing genome sequences from older sources (2014 & 2016). The genes highlighted in the KEGG database were the ones utilized within the analyses. Newer genomic analyses of these specific genomes (2021) highlight differing numbers of genes. The genomic differences can be seen in Table 1 and Table 2.

The re-sequencing of previous genomic sequences is a valuable resource, due to the improvements in sequencing technology. The new annotations to strain *L. reuteri* JCM 1112 give no reason as to why previous sequences had fewer genes than *L. reuteri* DSM 20016 or where the newly classified genes have come from. Potentially, this could be attributed to the re-annotation of the *L. reuteri* JCM 1112 strain when it was put through the newer pipeline.

There are other *L. reuteri* strains that may also offer human host health benefits. Some of these strains have undergone metabolic investigations through genome-scale models and genomic comparisons. The following research suggests that metabolomic research provides preliminary data on the usefulness of strains as probiotics, specifically those of the *Limosilactobacillus* genus. *L. reuteri* SD2112/ATCC 55730 and *L. reuteri* MM4-1A/ATCC PTA 6475 are both able to adhere to the human gut using mucus-binding proteins, along with producing vitamins and antimicrobial agents [35]. *L. reuteri* MM4-1A is a currently used probiotic that is used to prevent bone loss and to promote immunomodulation [36]. Genomic-scale models of another strain, *L. reuteri* KUB-AC5, highlighted its preferable growth environment, highlighting the use of transcriptome analysis to analyze metabolic routes and optimize cell growth [37]. This strain is highly competitive against *Salmonella* in poultry [38]. Further analysis of these strains and others, like *L. reuteri* DSM 20016, could provide valuable information as to the potential of these probiotics and their optimal growth conditions. The genomic data for these strains can be found in Table 3.

Analysis of the 1904 protein genes in *L. reuteri* DSM 20016 indicated 200 key genes with relation to human host health benefits. These highlighted genes were selected due to their ability to aid the probiotic bacterium in its survival and/or due to the metabolites’ known effects on human health. The desirable properties of a probiotic include its ability to tolerate acid and bile, adhere to epithelial surfaces, compete against enteric bacteria, have antimicrobial activity, and have bile salt hydrolase activity [39]. Because of this, any genes tied to acid resistance (GABA shunt), competition (reuterin), and adherence were selected. The microcompartment protein, propionate metabolism, and cobalamin synthesis genes were included as these are needed in *L. reuteri* DSM 20016 to produce reuterin and propionate. Probiotics can produce bacterial metabolites that can provide human host health benefits [40]. These metabolites include: SCFAs (acetate, propionate, and butyrate), which aid in gut integrity, glucose homeostasis, and immunomodulation [41]; histamine, which may be pro-inflammatory tumor necrosis factor (TNF)-suppressive [42]; amino acids; and vitamins [43,44]. These metabolites would need to be secreted, suggesting that the secretion pathways are also important. Genes for pyruvate metabolism, sugar uptake, and glycolysis were included, as these are needed for the formation of SCFAs.

These were located onto the genome using DNAPlotter as seen in Figure 3. The pathways and descriptions of these genes can be found in Appendix A. These genes account for around 9.63% of the genome. It also details the 12 areas that are involved in cellular processes and in potentially producing human host health benefits within the human host.

The main highlighted areas included the production of propionate, a short-chain fatty acid (SCFA), which involves the pdu operon and bacterial microcompartment proteins. The production of cobalamin is needed to synthesize reuterin through the same pdu operon. Sugar and pyruvate are needed to form short-chain fatty acids (SCFAs), such as propionate, lactate, and acetate. There is a potential pathway for butyrate production through a currently uncharacterized pathway. Because of this, genes for sugar uptake, glycolysis, pyruvate metabolism, and SCFA synthesis were investigated. 

The pathways within *L. reuteri* DSM 20016 are vital to understanding the function and production of this bacterium. Through the understanding of these pathways, *L. reuteri* DSM 20016 could be genetically or metabolically engineered as therapeutics for human diseases or as potential drug delivery systems, with synthetic probiotics currently being used in the targeting of cancer, treatment of infectious agents, and other functions [15]. Probiotics can also be designed and able to function as diagnostic tools for probing and identifying diseases, although more research will be needed to tailor these advances to increase the efficacy of disease therapy [45].

### 3.1. Pdu Operon: Propionate Metabolism, Reuterin, Cobalamin Synthesis and Microcompartments

Reuterin and short-chain fatty acids are the most important products of this pathway. Reuterin is an antimicrobial compound, consisting of isomers of 3-hydroxypropionaldehyde (3-HPA), that can inhibit enteric bacterial growth. It is synthesized when the bacterium is in contact with enteric microbes and causes oxidative stress by altering thiol group modification in *Escherichia coli* proteins [46]. Reuterin is produced from the fermentation of glycerol by an enzyme known as glycerol dehydratase/propanediol dehydratase.

The production of reuterin, an antimicrobial that targets enteric bacteria, is tied to cobalamin synthesis and propionate metabolism. The genes that encode these regions are largely confined to one cluster, known as the pdu operon. There were 54 genes linked to these pathways within *L. reuteri* DSM 20016, with the most (30) attributed to the synthesis of cobalamin. The cobalamin synthesis genes are all encoded in the pdu operon. A STRING map was created to showcase these genes, which can be viewed in Figure 4. The figure shows two clusters. The top cluster houses many of the cobalamin synthesis genes; the bottom cluster contains most of the genes for propionate metabolism and bacterial microcompartment formation.

The conversion of glycerol to reuterin uses propanediol dehydratase and a vitamin B12-cofactor. The enzyme is cobalamin-dependent, so the B12-cofactor is required in this reaction. This pathway can be seen in Figure 5, which proposes the mechanism in which the antimicrobial is formed. Propanediol dehydratase is composed of three subunits classified as pduCDE. PduCDE metabolizes glycerol to 3-HPA in resting *L. reuteri* cells [47]. There is an ideal ratio of glucose to glycerol for maximal 3-HPA formation, with excess glucose causing reduction of 3-HPA to 1,3-PD [48].

Propanediol dehydratase is the key enzyme in the formation of reuterin, but it is also involved in 1,2-propanediol (1,2-PD) degradation to propionate—a SCFA. Glycerol and/or 1,2-PD are required substrates for the reactions of pduCDE, which only occur when the cell is resting [Amin et al., 2013]. Propionaldehyde/propionate is formed if 1,2-PD is available; reuterin is formed if glycerol is available. Glucose increases 1,2-PD metabolism in *L. reuteri* DSM 20016 but is not essential, with glucose presence likely not influencing further propionate formation [49]. An absence of glucose in resting cells results in 3-HPA secretion [50]. This would suggest that lower levels of glucose are favored for this reaction.

Due to the toxicity of acrolein and propionaldehyde, it is likely that the pduCDE reactions occur within a microcompartment. The bacterial microcompartment is made up of nine genes within *L. reuteri* DSM 20016, including that of eutP, eutS, eutN, pduM, pduJ, pduK, pduB, pduA/eutM, and eutJ. The bacterial microcompartment can be seen in Figure 6 (dashed hexagon), wherein it encompasses the reaction responsible for propionate metabolism. The reaction seen in Figure 5 (reuterin production) also takes place in this bacterial microcompartment.

Vitamin B12 coenzyme is vital in the formation of both of these metabolites. KEGG details a clear path from L-glutamyl-tRNA (glu) and potentially other compounds through the anaerobic synthesis pathway, which features the insertion of cobalt. Both pathways can be observed in Figure 7. *L. reuteri* DSM 20016 has four cobalt transport proteins, located on the cob/cbi operon, which may provide the cobalt required for the 1,2-PD reaction and the synthesis of antimicrobial reuterin.

The end of the propionate metabolism pathway yields propionate, which may cause an anti-obesity effect by inducing hypophagic effects [51]. However, another study detailed that the SCFA induced hyperglycaemia in mice by increasing glucagon and impairing the action of insulin. In humans, propionate may lead to insulin resistance and obesity [52], as higher levels of propionate were found in overweight individuals in comparison to that of lean subjects [53]. While propionate is the second-most abundant SCFA produced by *L. reuteri* F275, *L. reuteri* DSM 20016 is a lab culture of F275, which may have undergone genomic changes due to lab cultivation. Gut-derived propionate may reduce cancer cell proliferation in areas outside of the gut, such as the liver, as propionate is often uptaken by this organ [54,55], or the lungs, by inducing cell apoptosis in lung cancer through sodium propionate treatment [56]. While more research might be needed on the effect of propionate on obesity, the potential anti-cancer effects could outweigh the risk.

### 3.2. Sugar Uptake

The uptake of glycerol and glucose is required for the formation of reuterin and other SCFAs, due to their requirement in glycolysis and in the maintenance of a favorable reuterin production. Utilizing KEGG, two genes were identified as glucose uptake proteins (Lreu_0032; Lreu_0418).

Glucose enhances glycerol metabolism and maintains a desirable NAD/NADH ratio, which favors an accumulation of 3-HPA. A molar ratio of 0.33 glycerol:glucose forms the most-favorable accumulation of 3-HPA, suggesting that both glucose and glycerol uptake is important for the formation of reuterin in *L. reuteri* DSM 20016 [57]. Glycerol uptake facilitator protein (Lreu_1752) and glycerol kinase (glpK; Lreu_1065) are possible genes required for the movement of glycerol into the cell and for its metabolism within the bacterium.

### 3.3. Pyruvate Synthesis and Glycolysis

Pyruvate is a predecessor for energy generation but is also the precursor for the production of many different SCFAs. While many different sugars can form pyruvate, *L. reuteri* DSM 20016 does have the mechanisms for lactate and glucose metabolism. There are four identified genes for D-lactate dehydrogenase and eleven currently detailed for pyruvate formation from glucose through the Embden–Meyerhof pathway (EMP).

*L. reuteri* DSM 20016 lacks the unique region located in *L. reuteri* JCM 1112, which encodes genes associated with glycolysis. Homologs for most of these genes were found within the *L. reuteri* DSM 20016 genome.

There are currently three unidentified genes within the glycolysis pathway: phosphofructokinase (EC:2.7.1.11) (EC:2.7.1.146) (EC:2.7.1.90), glyceraldehyde 3-phosphate dehydrogenase (EC:1.2.1.12) (EC:1.2.1.59), and phosphopyruvate hydratase/enolase (EC:4.2.1.11). These can be seen in Figure 8. Despite the extensive use of UniProt, these genes could not be located, suggesting that there may be another pathway being utilized or that another enzyme catalyzes these reactions. 

While KEGG suggests that the enzyme, bisphosphoglycerate mutase (EC:5.4.2.4), is not present within the cell, the conversion of glycerate-1,3-P2 is slowly catalyzed by phosphoglycerate mutase (EC:5.4.2.11) instead.

The predominant glycolysis pathway at the end of bacterial growth in *L. reuteri* DSM 20016 is the heterolactic pathway (EMP), which produces lactate and ethanol. However, in higher pH environments, the switch from the EMP pathway to the heterolactic phosphoketolase (PKP) pathway is most likely to prevent further increase in acid production and pH, despite yielding less lactate [58]. A STRING network map can be seen in Figure 9.

### 3.4. Histamine Production

Histamine is produced in *L. reuteri* DSM 20016 through histidine carboxylase (hdc)—an enzyme responsible for the conversion of L-histidine to biogenic amine histamine. In Gram-positive bacteria, this hdc is pyruvoyl-dependent. The genome is home to an hdc operon, consisting of hdcA (histidine carboxylase, pyruvoyl type; Lreu_1832), hdcB (hypothetical protein; Lreu_1831), and hdcP (arginine: ornithine antiporter/lysine permease; Lreu_1833). HdcB has an unknown function but is co-transcribed alongside hdcA as bicistronic mRNA. HdcC is located on the membrane and most likely is responsible for the transport of histidine and/or histamine [59]. A rsiR gene is believed to modulate histamine production in *L. reuteri* 6475, with a study detailing that TNF suppressive activity and a 97% to 100% identity hit exists within *L. reuteri* DSM 20016 [60]. The inactivation of the rsiR gene results in the decreased production of histamine by the bacterium. While the gene is proposed in *L. reuteri* DSM 20016, rsiR could not be identified during the search, as the genomic sequence was not publicly available.

Histamine regulates cells of the innate and adaptive systems, and this is dependent on histamine receptors [61]. *L. reuteri*-derived histamine suppresses TNF production by the activation of histamine receptor type 2 (H2) and MEK/extracellular MAP kinase signaling through TLR2-activated THP-1 cells [62]. The regulation of histamine is very important within *L. reuteri.* The rsiR gene reduces anti-inflammatory effects in vivo in mouse models of acute colitis induced with trinitrobenzene sulfonic. With an intact regulatory gene, colitis can be alleviated in mouse models [60]. However, the bacterial secretion of histamine may be beneficial in immunomodulation. In mice, gut-derived histamine influenced the immune response within the lungs, reducing lung eosinophilia and suppressing ovalbumin-stimulated cytokine secretion in ex vivo lung cells [63]. The STRING network for histamine production can be seen in Figure 10.

### 3.5. Bacterial Sec Pathway

The bacterial sec pathway is utilized in the exportation of proteins across the plasma membrane through the detection of a hydrophobic N-terminal leader sequence. Due to the pathway’s connection to translation, as seen in the STRING network map in Figure 11, it is likely that the sec pathway is responsible for many protein secretions and for the insertion of membrane proteins in Gram-positive bacteria [64]. This general protein secretion pathway may have ties to adhesion in *L. reuteri* DSM 20016 by allowing the incorporation of important cell surface proteins.

The translocase protein is made up of the secYEG pre-protein channel and ATPase motor secA. The secYEG protein can be associated with further Sec proteins, such as secDFYajC and other auxiliary proteins like yidC. In the general secretory pathway, secretion can be SRP-mediated or secB-medated, with the SRP-mediated pathway featuring secYEG and ftsY and the secB mediated pathway including secA, secYEG, and ftsY [65].

### 3.6. Adhesion and Colonization

Adhesion and colonization are largely unexplored within *L. reuteri* DSM 20016, however, information from related strains can provide an insight into this mechanism. Figure 12 details the key genes that were identified, which could be related to adhesion within this strain. Unlike the other areas researched within this paper, many of these genes were identified through reading related scientific papers and comparing genes from closely related strains. The initial step was to identify genes that had a known function of adhesion and colonization in related strains.

Sugars can be used in colonization, as supported by a study that found that glucosyltransferase A (gtfA) and inulosucrase (inu) in *L. reuteri* TMW 1.106 aid in cell aggregation and biofilm formation in the murine gut. While inulosucrase is not locatable within the bacterial genome, gtfA can be identified as sucrose phosphorylase (Lreu_1542), which uses sucrose-dependent auto aggregation [66]. This mechanism has already been utilized in studies featuring sucrose and maltose-loaded dextranomer microspheres (DM) in the treatment of NEC. *L. reuteri* formulates biofilms on the surface of the DM, reducing histological injury, reducing inflammation of the intestine, and increasing host survival in an animal model [67].

In the murine gut, a high-molecular-mass surface protein (lsp) and methionine sulfoxide reductase B (msrB) contribute to the performance of adhesion in *L. reuteri* 100-23. While the lsp cannot be located within the genome, the study does suggest that a lsp homolog is present within *L. reuteri* DSM 20016. Msr reverses the loss of biological activity of proteins due to methionine sulfoxide oxidation and other oxidative damage [68]. MsrA can aid in the maintenance of adhesins in bacterial cells [69]. MsrB may be required to lessen nitric oxide produced by cells, which may provide protection from pathogenic bacteria [70].

Etzold et al., (2014) [71] detail an adhesive protein that binds to mucus and features six repeats, a YSIRK signal sequence, and LPXTG motif. While produced by *L. reuteri* DSM 20016 and showing intermediate binding, the Lar_0958 homolog is not reflected in the genome. A discovered mucus adhesion protein (mapA) is a rather important cell adhesion factor found within bacteria and located within the genome (Lreu_0296) utilizing Blast similarity search (https://www.genome.jp/tools-bin/search_sequence (accessed on 15 June 2020)). MapA can bind to mucus and also to Caco-2 cells [72]. There is evidence through literature [73], string, and genbank accession AJ293860 that indicates that mapA may be linked to three other closely situated genes (Lreu_0293; Lreu_0294; Lreu_0295), which are related to L-cystine transport and to a sequence of genomic DNA (ORF1), which is located between Lreu_0292 and Lreu_0293. The literature also detailed that a knock-out mutant of MapA resulted in reduced persistence in the murine gut, suggesting there are other adhesion factors occurring simultaneously. One of these could be sortase A (srtA), which, when mutated resulted in reduced adhesion to mucus and Caco-2 cells [74], and fibronectin-binding proteins could also aid in binding to epithelial cells [75]. Another potential factor could be the inclusion of D-alanyl esters into the teichoic acids of the cell wall through the dlt operon. This is important for acid tolerance, cAMP resistance, and adhesion [76]. In *L. monocytogenes*, the dlt operon is required for virulence and adhesion [77].

EF-Tu and GroEL are proteins with the potential to modulate adhesion. Ef-Tu is an elongation factor that aids in the binding of aminoacyl-tRNA to ribosomes for translation. It is involved in immune evasion, immune modulation, and in binding to fibronectin and extracellular matrix proteins, facilitating the adhesion process [78]. GroEL is a heat shock protein that has been implicated in the cell attachment and immune modulation processes. In *Lactobacillus johnsonii* NCC533 (La1), this protein stimulates the immune response and aggregates *H. pylori* while also binding to mucins and epithelial cells found in the gut [79]. The La1 strain was also found to bind to human intestinal cells and mucin through EF-Tu proteins, suggesting that other lactic acid bacteria may also have this property [80].

GroEL and EF-Tu are often found intracellularly, with evidence of these proteins appearing on the cell surface needed to determine if *L. reuteri* DSM 20016 has the binding capability. This could be done through enzyme-linked immunosorbent assay and immunoblotting.

The second step was identifying proteins that extended the membrane (transmembrane) and were located on the outside of the cell (extracellular). These proteins could potentially have adhesion and colonization properties. A previous study published in 2003 by Wall et al. [81] used phage display technology to reveal 52 transmembrane and extracellular proteins from *L. reuteri* DSM 20016. Due to the age of the study, these proteins were not labeled with their appropriate locus tag, meaning BLAST Sequence Similarity Search was needed to identify the most likely candidate genes. Most of the sequences reported in this study were not complete matches to the current genome, so an identity of >90% was used, meaning 90%+ of the characters matched between the two sequences. The interpreted data can be found in Appendix A, wherein, in some cases, no likely genes were located for proposed proteins. Figure 13 features these identified genes in a STRING network. More research will be needed to identify the role of these proteins and whether they are involved in adhesion, signaling, or other roles.
Figure 12STRING network map of the key adhesion/colonization factors proposed within *L. reuteri* DSM 20016 (medium confidence, 0.400). The genes identified here had known adhesion and colonization functions (https://string-db.org/; (accessed on 23 April 2022) version 11.5) [82].
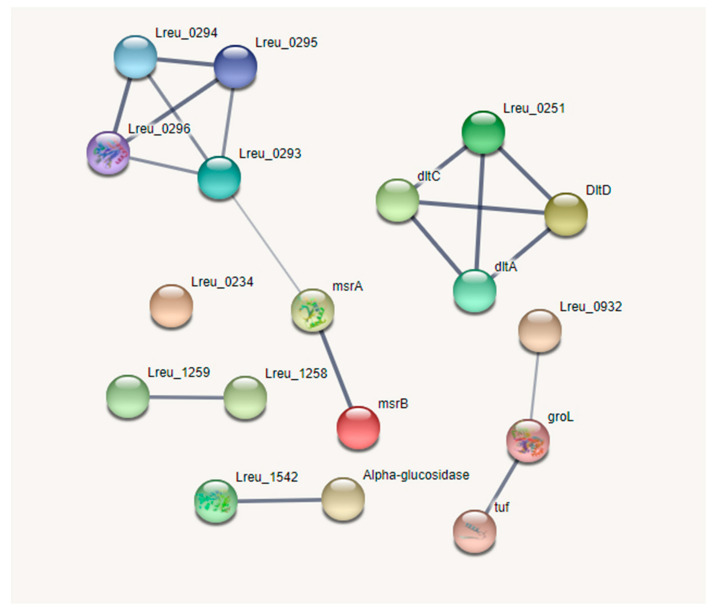



### 3.7. Short-Chain Fatty Acids

Short-chain fatty acids (SCFAs) are important in protecting the gut from enteric pathogens due to their antimicrobial properties and ability to, with difficulty, cross the membrane and affect intracellular pH. Even subinhibitory levels of SCFA can affect the motility of salmonella strains and affect their biofilm formation and gene expression [83]. A study by Kahouli et al. (2015) [84] conducted on *L. reuteri* strains measured SCFA production, including in strain *L. reuteri* NCIMB 11951—an alternative collection code for DSM 20016. This study found that strain *L. reuteri* NCIMB 11951 produced significantly more SCFAs than other *L. reuteri* strains and also showed the highest survivability out of all the bacterial strains tested after 8 h of incubation. *L. reuteri* NCIMB 11951 produces a high amount of acetic acid (131.2 ± 4.8 mg/L or ~2.185 mM) and propionic acid (111.2 ± 5.5 mg/L or ~1.501 mM). There was a significant production of butyrate (28.6 ± 4 mg/L or ~0.3246 mM). While not a SCFA, there was a medium amount of lactic acid (369.1 ± 15.1 mg/L or ~4.097 mM) formed, as well. The study also concluded that, for 48 h of treatment, this particular *L. reuteri* strain reduced colon cancer growth by 32.02 ± 0.97%. Propionate and butyrate may be the key factors, as they can induce differentiation and induce autophagy in human colon cancer cells, providing an anticancer effect [85].

Lactic acid bacteria, such as that of *L. reuteri* DSM 20016, produce SCFAs from carbohydrates. Figure 14 proposes the genes responsible for SCFA production in *L. reuteri* DSM 20016. *Limosilactobacillus* strains are a major factor in the formation of lactic acid/lactate. Lactate derived from the microbiota can help to accelerate the regeneration of epithelial cells through G-protein-coupled receptor 81 [86]. KEGG lists six genes responsible for the conversion of pyruvate to L-lactate within *L. reuteri* DSM 20016 as L-lactate/malate dehydrogenase (NAD). Pyruvate to D-lactate features four genes using the enzyme D-lactate dehydrogenase.

Acetate formation within *L. reuteri* DSM 20016 is also largely explored (Figure 15). Pyruvate is converted to acetyl-CoA through pyruvate dehydrogenase. Within Gram-positive bacteria, the pyruvate dehydrogenase subunits require cofactors, including coenzyme A, FAD, NAD+, and lipoate. There are mechanisms for the formation of these cofactors, with coenzyme A being synthesized from pantothenic acid and lipoate from octanoate. Acetyl-CoA is converted to acetyl-P through phosphate acetyltransferase, then finally converted by acetate kinase to acetate. Acetate may be able to fine-tune the inflammatory response in reference to gout caused by monosodium urate crystals in mice by inducing the caspase-dependent apoptosis of neutrophils [87]. In the intestine, acetate can enhance the production of Th1 and Th17 T-effector and regulatory cells by producing certain cytokines, such as IL-17 and IFN-y/IL-10. However, there was a decrease in anti-CD3-induced inflammation [88].

Gut-derived butyrate regulates the immune system, correlating with the concentration of T-regulatory cells within the colon and enhances the intestinal barrier by aiding in the assembly of tight junctions through AMP-activated protein kinase [89,90].

There are two identified pathways for butyrate formation in *L. reuteri* DSM 20016, which can be observed in Figure 16. The first is through butanoate metabolism, which, when referring to KEGG, is incomplete. Utilizing BlastP, UniProt, and literature detailing these enzymes in related *Levilactobacillus brevis* [91], possible genes of the pathway can be located. The potential genes include acetyl-CoA acetyltransferase (Lreu_0052), 3-hydroxyacyl-CoA dehydrogenase (Lreu_0103), and 4-hydroxybutyryl-CoA dehydratase (Lreu_0887). The last two steps are rather unexplored. The gene responsible for conversion of crotonyl-CoA to butyryl-CoA may be that of Lreu_0981, according to BlastP; however, this gene is also found in a similar reaction requiring acyl carrier proteins (ACP). The final conversion of butyryl-CoA to butanoate/butyric acid is not currently clear, as there are no potential enzymes of the suggested pathways.

A pathway that is completed within *L. reuteri* DSM 20016 is that of the currently uncharacterized FAS II pathway, which forms long-chain fatty acids, including that of butyryl-[acp]. The FAS II pathway uses holo-carrier proteins for the acyl-activation of fatty acid biosynthesis and also Fab proteins, which catalyze the change to butyryl-[acp]. The conversion of the acyl-proteins to the carboxylate may be formed through the use of a thioesterase, such as that of Lreu_0335. Another thioesterase, potentially, tesB is tied to butyrate formation, but removal only results in a slight decrease in butyric acid formation, suggesting there is more than one mechanism [92]. More research on this pathway will be needed, however, from studies done on the *L. reuteri* DSM 20016 strain, to investigate how butyrate is formed.

### 3.8. Vitamins

*Limosilactobacillus reuteri* can produce many different types of vitamins, with specific strain DSM 20016 known to produce folate and cobalamin [28,93]. Through analysis of the genome, the strain has the mechanisms to synthesize folate and riboflavin and the ability to utilize thiamine, pantothenic acid, niacin, and biotin. The genes identified in these pathways have been mapped in STRING in Figure 17 and Figure 18.

Folate (B9) is synthesized by intestinal bacteria from GTP, erythrose 4-phosphate and phosphoenolpyruvate. The most important function of folate is its necessity as a cofactor for the synthesis of purine and thymidine nucleotides. Vitamin B9 is also associated with the immune system as a survival factor for T-regulatory cells, which express folate 4 receptors [94]. A natural source of folate could be beneficial as synthetic folic acid supplements, taken in excess, can result in an increased colorectal and prostate cancer risk [95,96]. On the other hand, deficiencies in folate or low folate status can lead to an increased cancer risk [96]. In *L. reuteri* DSM 20016, the folate pathway can be followed from 7,8-dihydroneopterin, utilizing 4-aminobenzoate in the formation of DHF and then reducing to folate by dihydrofolate reductase (Lreu_0770) (Figure 19).

Multi-domain polypeptides utilizing biotin (Vitamin H) form acetyl-CoA carboxylase utilized in the formation of malonyl-CoA from acetyl-CoA using ATP. This reaction is required in the formation of fatty acids using the FAS II pathway, which can form butyric acid and other long-chain fatty acids. In its absence, mice will have reduced body fat due to continuous fatty acid oxidation [97]. The biotin-dependent enzyme requires biotin carboxylase (Lreu_0984), carboxyltransferase (Lreu_0982; Lreu_0983), and biotin carboxyl carrier protein (Lreu_0984). Biotin is attached to biotin carboxyl carriers through biotinylation and through the use of birA, which is identified as Lreu_0734 in *L. reuteri* DSM 20016 [98]. Synthesis within *L. reuteri* DSM 20016 has not been explored, but the bacterium does contain several transport proteins, which suggests that biotin may be exported into the cell for fatty acid synthesis.

Humans require exogenous sources of thiamine (vitamin B1) due to an inability to synthesize this vitamin within the body. Thiamine is involved within energy metabolism and as a cofactor, with thiamine-dependent enzymes required for neurotransmitter synthesis. Thiamine deficiency can result in beriberi, peripheral neuropathy, and cardiomyopathy [99]. The pathway for thiamine synthesis within *L. reuteri* DSM 20016 is largely unexplored; however, the formation of thiamine pyrophosphate (TPP) from thiamine is documented, utilizing the enzyme thiamine phosphate phosphatase (Lreu_1168) (Figure 20).

There is likely a carrier-mediated mechanism for the uptake of TPP by colonic cells, and it is that this phosphorylated compound may be used as cellular nutrition or within the body as a cofactor [100]. TPP can be used to decarboxylate pyruvate for the pyruvate dehydrogenase complex and can cleave alpha-keto acids [101]. If the pathway is explored, any potentially synthesized thiamine could help support meeting the dietary requirement and provide essential cofactors for human-based reactions.

Riboflavin (vitamin B2) produces flavin adenine dinucleotide (FAD) and flavin mononucleotide (FMN), which are required as cofactors in energy metabolism and other oxidative and reductive reactions. Induced through GTP and 5-ribulose phosphate, the pathway for the synthesis of riboflavin is found within *L. reuteri* DSM 20016 (Figure 21). FAD is required for fatty acid oxidation through acyl-CoA dehydrogenase and is needed as a coenzyme in pyruvate dehydrogenase to form acetate. RF-enriched foods are used in a biotechnological application utilizing lactic acid bacteria to produce vitamin-enriched food, such as using *L. planetarium* for RF-enriched sourdough bread and pasta [102]. Riboflavin has many health effects, including reducing hepatocellular injury in mice with liver ischaemia by exhibiting anti-inflammatory and antioxidant properties [103]. Other benefits include providing protection against oxidant-mediated acute lung injury [104]. The displayed antioxidant effects could be useful in the treatment of other inflammatory diseases.

While the mechanism for the synthesis of niacin (vitamin B3) in *L. reuteri* DSM 20016 has not been described, the formation of nicotinamide and NAD from vitamin B3 is detailed in Figure 22. Niacin may reduce the severity of colitis and the colonic level rise of TNF-a, VEGF, angiostatin, and endostatin induced by iodoacetamide, while also providing normalization to IL-10 levels within the colon [105]. This may make niacin-producing microbiota effective against iodo-acetamide-induced colitis by altering inflammatory changes through the GPR109A receptor. Nicotinamide adenine dinucleotide (NAD) and NAD+ are required as cofactors for proteins that regulate metabolism and are used in many oxidative and reductive reactions. Homeostasis of NAD+ is required for longevity and an improved lifespan, with alterations to the levels resulting in potential pathologies [106]. In the case of niacin deficiencies, effects can include the skin, gastrointestinal symptom, and the neural pathway. If niacin is found to be synthesized in *L. reuteri* DSM 20016, the effect of bacterial-derived niacin could be evaluated, potentially offering these anti-inflammatory effects and helping to meet dietary requirements to prevent niacin deficiency and pellagra.

Coenzyme A is required in the formation in some SCFA, such as acetate, and for the formation of the B12 coenzyme needed for reuterin production. In *L. reuteri* DSM 20016, coenzyme A is derived from pantothenic acid. The KEGG pathway for this gene details a pathway of 5 genes between (R)-pantothenate (vitamin B5) to coenzyme A utilizing cysteine or a derivative (Figure 23). In bacteria, pantothenate is synthesized from L-valine and L-aspartate; these pathways do not exist within *L. reuteri* DSM 20016. This suggests that pantothenate is imported from the gut for coenzyme A synthesis.

### 3.9. Amino Acids

Amino acid utilization could be an important next step in regulating energy and protein homeostasis, as bacteria within the gastrointestinal tract are important in dietary metabolism. With further research, probiotics may be able to modulate amino acid utilization within the gut [107]. The bacterial composition within the porcine gut reflected the synthesis of essential amino acids. By incorporating 15N in lysine and 14C into amino acids, the effects of the microflora in the synthesis of amino acids were measured in the: reporting absorption of 14C-labeled valine, isoleucine, leucine, and phenylalanine. While lysine had some 14C labeling, the predominant amount of 15N-labeled lysine was estimated at 1.3 g/d. The evidence from this study does suggest that some of the amino acid requirements may be met by the porcine microflora in the ileum [108]. Another study conducted on the gastrointestinal tracts of humans found that the small intestine is largely responsible for microbial lysine uptake, with around 1–20% of the plasma, urinary, and protein lysine being derived from the microbial sources of the gut through the use of 15N labeling [109]. The net contribution of microbial amino acids to dietary requirements cannot be stated as of yet; however, future research into the area could provide new insight into microbially-derived amino acids and their usages within the body.

Lysine is produced within *L. reuteri* DSM 20016, with a mostly complete KEGG pathway, from L-aspartate. This pathway can be seen in Figure 24. Lysine is not produced in humans and must be acquired through diet. It is required in protein synthesis and in the production of certain short-chain fatty acids. Due to the higher lysine requirements for late-stage pregnancies, lysine-fortified foods and more understanding of gut-derived lysine may be ways to counteract lysine deficiency, which can cause stress-induced anxiety in rats [110,111]. Lysine-fortified rice improved growth performance and lysine availability in rats, while lysine supplementation in a study in Ghana resulted in decreased diarrhea morbidity in children and respiratory morbidity in men [112,113]. A total of 30–42% of lysine released from dietary sources was metabolized in humans, suggesting that the microbiota may play a role in supplementing lysine requirements in the host [114].

Notably Figure 25 featured two genes, Lreu_1544 (dipeptidase) and coaE, that were not connected to the other genes within the STRING network in version 11, even when lowering the confidence. However, in the current version (version 11.5), these genes are connected through medium confidence. This was potentially due to unconnected pathways within the database, which have since been corrected. This highlights the importance of reviewing past genomics research, as many databases can be updated and new connections can be made.

While reviewing this pathway through UniProt, it was noted that gene Lreu_0613 was labeled as 2,3,4,5-tetrahydropyridine-2,6-dicarboxylate N-succinyltransferase in the KEGG database but as 2,3,4,5-tetrahydropyridine-2,6-dicarboxylate N-acetyltransferase in the UniProt database. The almost identical strain *L. reuteri* JCM 1112 had an identical amino acid sequence of the protein (Lar_0593) but also labeled the protein as a N-acetyltransferase. If we utilize the protein suggested by the UniProt database, we can assume that lysine may be synthesized through the acetyl-DAP pathway. The ability to synthesize lysine may make *L. reuteri* DSM 20016 useful in supplying the dietary requirements for the amino acid in the host.

There was also a pathway for pantothenate degradation to coenzyme A. The degradation of this vitamin is useful for the formation of the B12 cofactor needed in propionate and reuterin generation. 

### 3.10. Gamma-Aminobutyric Acid

There exists a bidirectional communication between the gut and brain, known as the gut–brain axis (GBA). Neural, endocrine, immune, and humoral signaling form the interaction of the GBA, with disruptions in these complex interconnections leading to dysbiosis and associated central nervous disorders, such as anxiety and depression [115].

Bacteria are known to both consume and/or produce a variety of neurotransmitters [116]. These neurotransmitters were searched for in the *L. reuteri* DSM 20016 genome. There was no detection of serotonin, melatonin, and acetylcholine. There was production of histamine, which has been previously discussed.

Neurotransmitters were searched for in KEGG and through using a metabolic reconstruction of *L. reuteri* JCM 1112 [117]. The unique regions do not encode genes responsible for neurotransmitter production, suggesting that the produced metabolites would be similar to that found in *L. reuteri* DSM 20016. The reconstruction found no presence of serotonin, melatonin, or acetylcholine in the metabolome.

When analyzing the pathways within *L. reuteri* DSM 20016 using KEGG, it is unlikely that nitric oxide is formed through nitrate metabolism, as *L. reuteri* DSM 20016 lacks these genes, due to it missing unique region II. Hydrogen sulfide is a known precursor required in the formation of cysteine from serine, with the enzyme required for this step (cysteine synthase) being located within the genome at Lreu_1553 (acetylserine-dependent), and Lreu_1792. *L. reuteri* DSM 20016 likely does not produce hydrogen sulfide, as the enzymes responsible for this (cystathionine γ lyase, cystathionine β synthase, and 3-mercaptopyruvate sulfurtransferase) are not locatable within the genome. Instead, *L. reuteri* is deemed to be sensitive to hydrogen sulfide concentrations [118].

Gamma-aminobutyric acid (GABA) was a metabolite found within a metabolic reconstruction of *L. reuteri* JCM 1112—a strain also derived from F275 [119]. GABA is an amino acid derived from glutamate that is found within plants, animals, and microorganisms. It is known as an inhibitory neurotransmitter within the brain and a metabolite that could influence the host physiology through the gut–brain axis, such as evidence that higher levels of faecal *Bacteroides* are negatively correlated with depression-related brain signatures [113]. GABA is known to be produced by lactic acid probiotics, although gut bacteria can produce and/or utilize GABA [120].

However, as seen in Figure 26, the pathways surrounding GABA and the GABA shunt are largely uncharacterized in *L. reuteri* DSM 20016. More characterization would be needed to determine how GABA is formed and utilized. Either GABA is produced through a currently unknown mechanism, or this strain uses GABA from the gut in bacterial processes. *L. reuteri* 100-23 is known to produce GABA—a process that generates acid resistance. Yet comparing the gene sequences of glutamate decarboxylase, which is the enzyme that catalyzes the conversion of glutamate to GABA (4-aminobutanoate), showed no results in *L. reuteri* DSM 20016.

The GABA shunt is known to bypass two steps of the TCA cycle, suggesting that not all enzymes in the TCA cycle are needed [115]. GABA could be produced through a characterized GABA shunt found in *Listeria monocytogenes*. The pathway in this strain has incomplete gene gaps between malate and oxaloacetate and between α-ketoglutarate and succinate [121]. This is similar to that of the *L. reuteri* DSM 20016 and suggests that the whole TCA cycle is not required for succinate biosynthesis. The *L. monocytogenes* strain only had two main enzyme genes, GABA aminotransferase and succinic semialdehyde dehydrogenase, both characterized in the *L. reuteri* strain (Lreu_0199 and Lreu_0034, respectively).

As in *L. monocytogenes*, this pathway likely gives *L. reuteri* DSM 20016 a system to cope with acid stress, making it a more effective probiotic, as it would have higher survival in acidic conditions. STRING analysis can be seen in Figure 27.

### 3.11. Limitations and Knowledge Gaps

Within the genome of *L. reuteri* DSM 20016, there are certain missing genes required to complete some pathways. Glycolysis is missing three unidentified genes (phosphofructokinase, glyceraldehyde phosphate dehydrogenase, and phosphopyruvate hydratase/enolase), which are required to complete the pathway. It could be possible that other pathways are utilized in these gaps; however, more research will be needed to identify the exact pathway and genes required. The limitations of this study are that there are likely other genes and pathways that were not located, which could provide health benefits to the host. Potentially reviewing this research with more up-to-date databases, and using in vitro and in vivo analysis of the produced metabolites of *L. reuteri* DSM 20016 could allow the quantification and identification of these key proteins.

Perhaps the most critical limitation would be the lack of investigation into any negative side effects of *L. reuteri* DSM 20016. *L. reuteri* DSM 20016 is known to form several antimicrobial genes, including those involved in the beta lactam, vancomycin, and cAMP resistance pathways. This strain is known to transport hemolysin (Lreu_1790 and Lreu_1789) and contain a hemolysin (hlyIII; Lreu_0771). However, while further research suggests beta lactam antibiotic resistance genes cannot be horizontally transferred due to the lack of plasmids and that hemolysins may be involved in bile stress, there will need to be a full investigation into the safety of this strain before it could be used within the human host [122,123]. Another interesting finding was the presence of clumping factor A (Lreu_0082), which is proposed to protect against the invasion of enteric bacteria through competitive exclusion [124].

A colicin V production gene is located within *L. reuteri* DSM 20016; however, due to the lack of plasmids, the bacterium most likely does not produce the peptide antibiotic. This is due to producing strains utilizing plasmids in the exporting process of colicin. A study highlights the use of a replacement of the leader peptide of colicin with that of a bacteriocin (due to their similar exportation styles) for transportation through the SEC pathway. The lactococcus used within this study did feature transformed plasmid pNZ-LR-ColV, potentially suggesting that this may not be feasible within this *Limosilactobacillus* strain. More research on this topic would be needed to identify whether *L. reuteri* DSM 20016 is capable of colicin V production and transportation.

More research on amino acids produced by this strain is needed, as there is little research on amino acids within *L. reuteri*. With the use of genomic analysis and experimentation, these amino acid pathways could be mapped and tested to identify whether these amino acids are incorporated or provide any benefits to the host. 

During STRING analysis, it was found that many pathways did not have significant or confident transcriptional factors, suggesting that there may be missing genes within these sequences. It could also be possible that some transcriptional regulators may not be marked within the genome. Identifying these transcriptional regulators will allow more understanding of the pathways within *L. reuteri* DSM 20016 and would allow the ability to amplify the production of beneficial compounds or repress the production of harmful compounds through genetic engineering. In this instance, the regulation of histamine could allow over-expression in cases of infection or repression when treating inflammatory diseases.

## 4. Conclusions

The ability of *L. reuteri* DSM 20016 to produce compounds greatly influence the immune system within the gut, and other parts of the body may make it viable in the treatment of certain inflammatory diseases and cancers. *L. reuteri* DSM 20016 is able to produce reuterin through the fermentation of glycerol using the same enzyme used in propionate metabolism (glycerol dehydratase) and other genes located in the closely linked pdu operon. Reuterin is an antimicrobial compound that makes this potential probiotic competitive against the enteric pathogens of the gut, as it is known to induce oxidative stress in *E. coli* species. The production of reuterin occurs in a bacterial microcompartment due to the toxicity of certain substrates. *L. reuteri* DSM 20016 is a natural inhabitant of the gut, and it has the genes required to allow it to bind to human Caco2 and epithelial cells. These genes include that of gtfA, srtA, msrB, mapA/Cnb, and fbpA. There is some evidence to support the idea that Ef-Tu and GroEL may also modulate adhesion by binding to fibronectin and mucins. Probiotics need to adhere well to the gut lining so that health-promoting metabolites are able to be secreted. They also need to have good resistances, with *L. reuteri* DSM 20016 showing evidence of this through its dlt operon (cAMP resistance and acid resistance) and through the GABA shunt (acid resistance). There is some presence of genes related to beta-lactam and vancomycin-resistance pathways, yet these need to be further explored. Beta-lactam resistance is unlikely to be horizontally transferred, as the strain lacks plasmids. Probiotics must be safe for the host, so more investigations will be needed to determine whether *L. reuteri* DSM 20016 produces any harmful products.

Vitamin B12 coenzyme is needed in the formation of both propionate and reuterin, with the anaerobic pathway of cobalamin synthesis being well characterized within this strain. These pathways require a good source of energy and metabolites, particularly those formed through glycolysis. Genomic analysis suggests that *L. reuteri* DSM 20016 uses the heterolactic pathway (PKP) for the generation of lactate and ethanol, although there are genes missing within both potential pathways. This suggests that the genes for the enzymes are uncharacterized or that the mechanism by which glycolysis occurs is not yet identified. Lactate is a microbial metabolite that is often produced by lactic acid bacteria, and it could be a key substance in gut health. A prior study also suggested that *L. reuteri* DSM 20016 produces high amounts of both acetic and propionic acid through their regular pathways. Butyrate is also produced by this strain; however, the pathway for this SCFA is very incomplete. A proposed pathway utilizing FASII and a thioesterase is a viable production mechanism, which has been investigated in other genera and species and is largely complete in the genome of this strain. If these SCFAs are secreted, then these could be beneficial metabolites in immunomodulation and in enhancing the intestinal barrier.

*L. reuteri* DSM 20016 has a functional secretory pathway through the bacterial Sec pathway. The pathway allows the addition of proteins to the cell membrane and the export of most proteins. In *L. reuteri* DSM 20016, the secYEG channel likely associates with yajC and yidC. SRP-mediated export is more likely due to the absence of secA in the genome, but the quantification of sec proteins will be needed to propose which mechanism exportation is conducted by. The strain has the genomic capacity to synthesize folate and riboflavin, which are known to influence the immune system and alter the inflammatory response when exogenous in the human gut. Exploring whether these vitamins are produced and exported in the secretome would be a good investigation to determine whether *L. reuteri* DSM 20016 could be used in food supplementation or direct dietary intake to supplement key vitamins in the human gut.

The *L. reuteri* strain has the hdc operon within its genome, which is composed of hdc-pyruvoyl-dependent hdcA and hdcB. RsiR is a proposed regulatory protein of this pathway, but, while a previous study suggested that the gene has an identity of 97–100% in this strain, the gene could not be localized on the genome in this study, as the genomic sequence for the gene could not be found in the literature.

The optimization of this strain through the metabolic engineering of the key metabolites identified in this atlas could aim to further improve this strain as a probiotic treatment for certain diseases. With more research into the produced amino acids and insights into its effect on specific inflammatory diseases and other pathologies, *L. reuteri* DSM 20016 could be an advantageous addition to the human microbiota. This strain could potentially complement other *L. reuteri* or other probiotics strains to provide a whole host of benefits. With an ever-expanding industry (expected to be worth USD 77 billion by 2025), research into probiotics could be beneficial for the local and national economy, creating job opportunities and potentially reducing the environmental effects (such as drug resistance and pollution) caused by the pharmaceutical industry. Using probiotics as a preventive measure for certain diseases could potentially save money spent on treatment. Similarly, more research into probiotics as drivers for increased drug efficacy and effective drug transport systems for administration could place the probiotics industry in line with other healthcare areas.

## Figures and Tables

**Figure 1 microorganisms-10-01341-f001:**
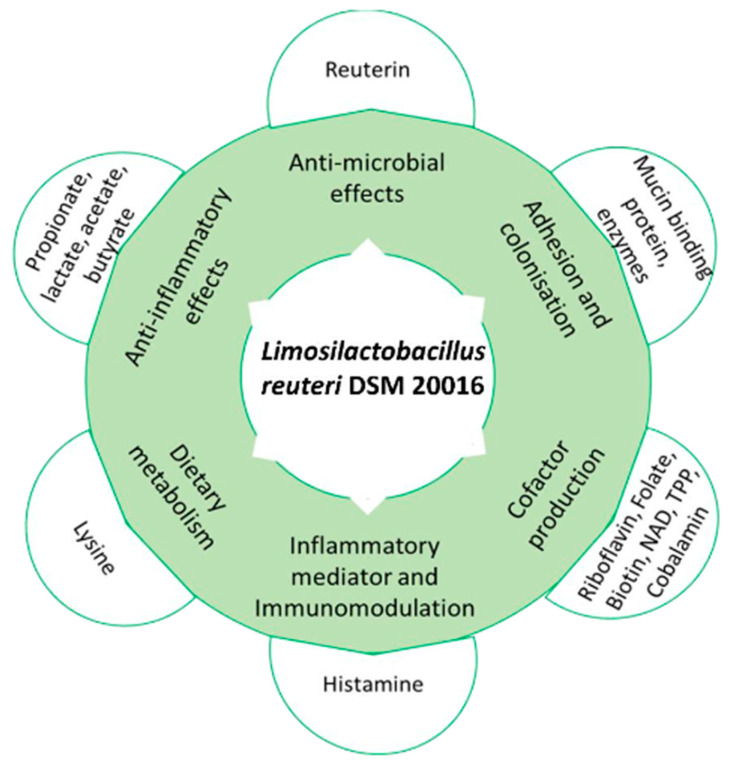
Flow chart of the health benefits and features of *L. reuteri* DSM 20016. The chart reports the effects of histamine, reuterin, and adhesion factors and labels certain products/metabolites that may provide health benefits within the host.

**Figure 2 microorganisms-10-01341-f002:**
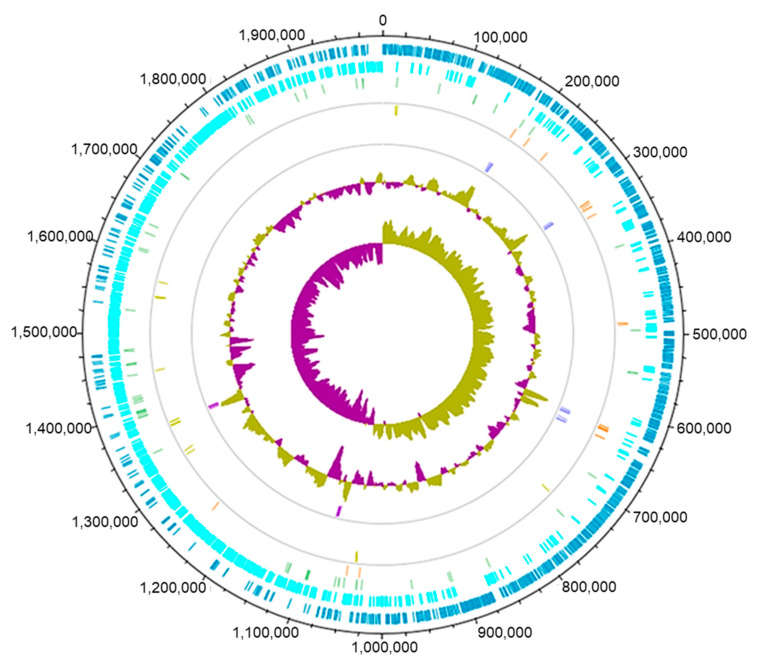
Genomic map detailing the chromosome of *L. reuteri* DSM 20016. From the outside circle in: Dark blue—coding sequence (CDS), forward track; light blue—CDS, reverse track; light green—tRNA, forward; orange—tRNA, reverse; yellow—rRNA, forward; purple—rRNA, reverse. GC plot—fraction of bases that are G or C. GC skew—degree at which GC content is skewed toward G or C. Made with DNAPlotter [34].

**Figure 3 microorganisms-10-01341-f003:**
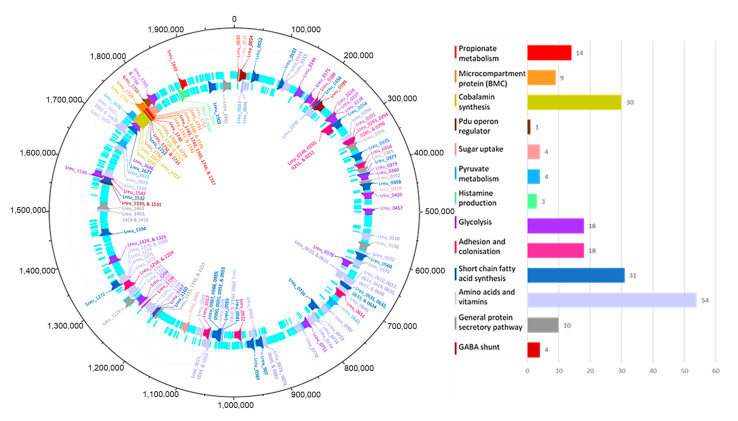
Genomic map of *L. reuteri* DSM 20016 detailing relevant identified genes tied to potential metabolites that could impact human health. Outer light blue ring—reverse coding sequence; inner light blue ring—forward coding sequence. Map shows color-grouped genes—labeled with the locus tags found in *L. reuteri* DSM 20016. Groups and colors include propionate metabolism—red, microcompartment protein (BMC)—orange, cobalamin synthesis—yellow, pdu operon regulator—dark brown, sugar intake—pink, pyruvate metabolism—light blue, histamine production—light green, glycolysis—purple, adhesion and colonization—dark pink, short-chain fatty acid synthesis—dark blue, general protein secretory pathway—gray, and GABA shunt cycle—light brown. Made using DNAPlotter [34].

**Figure 4 microorganisms-10-01341-f004:**
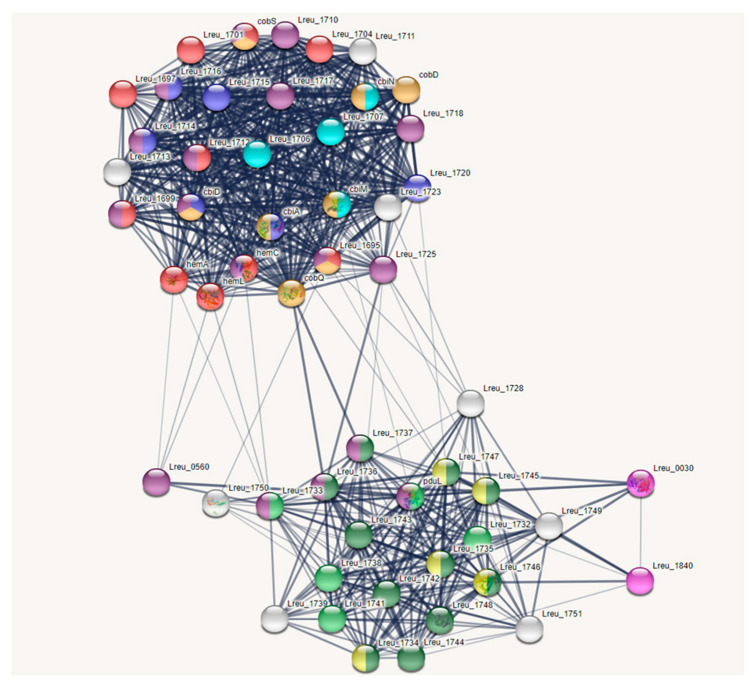
STRING network map of the pdu operon within *L. reuteri* DSM 20016, organized by KEGG pathways, UniProt keywords, and network clusters: dehydratase subunits and BMC (bacterial microcompartment) proteins—dark green, porphyrin biosynthesis and porphyrin metabolism—red, porphyrin and metabolism—blue, BMC domain and EutN/Ccml superfamily—light green, propionate metabolism—yellow, iron containing alcohol dehydrogenase—light purple, ABC transporters—light blue, cobalamin biosynthesis—orange, transferase—brown. (https://string-db.org/ (accessed on 10 September 2020)) [33].

**Figure 5 microorganisms-10-01341-f005:**
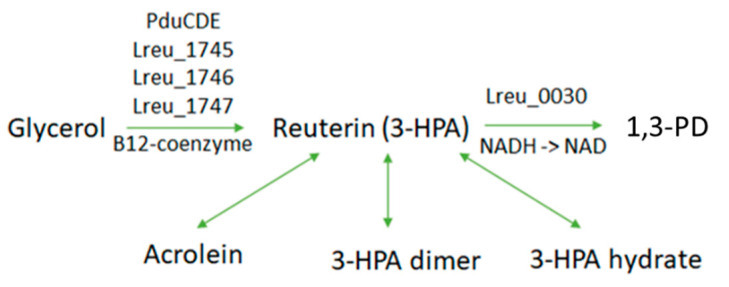
The production of reuterin through pduCDE/propanediol dehydratase and B12-coenzyme cofactor. Conversion of 3- HPA to 1,3-propanediol (1,3-PD) can occur via 1,3-propanediol dehydrogenase. The reuterin system encompasses 3-HPA, acrolein, its dimer, and its hydrate.

**Figure 6 microorganisms-10-01341-f006:**
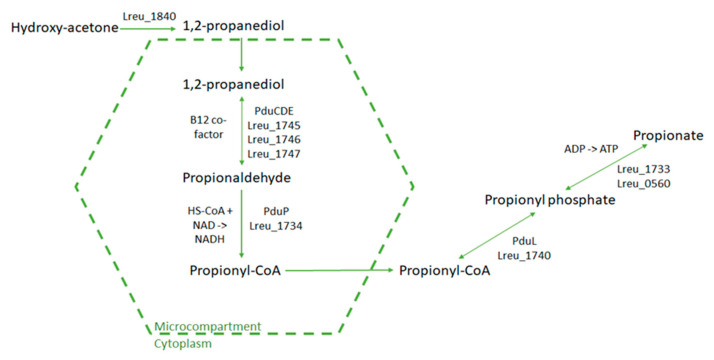
Diagram featuring the bacterial microcompartment within *L. reuteri* DSM 20016, including all relevant reactions of genes located in the pdu operon. The product of this reaction is the SCFA propionate. The pduCDE enzyme is propanediol dehydratase, which is used as glycerol dehydratase in the conversion of glycerol to reuterin.

**Figure 7 microorganisms-10-01341-f007:**
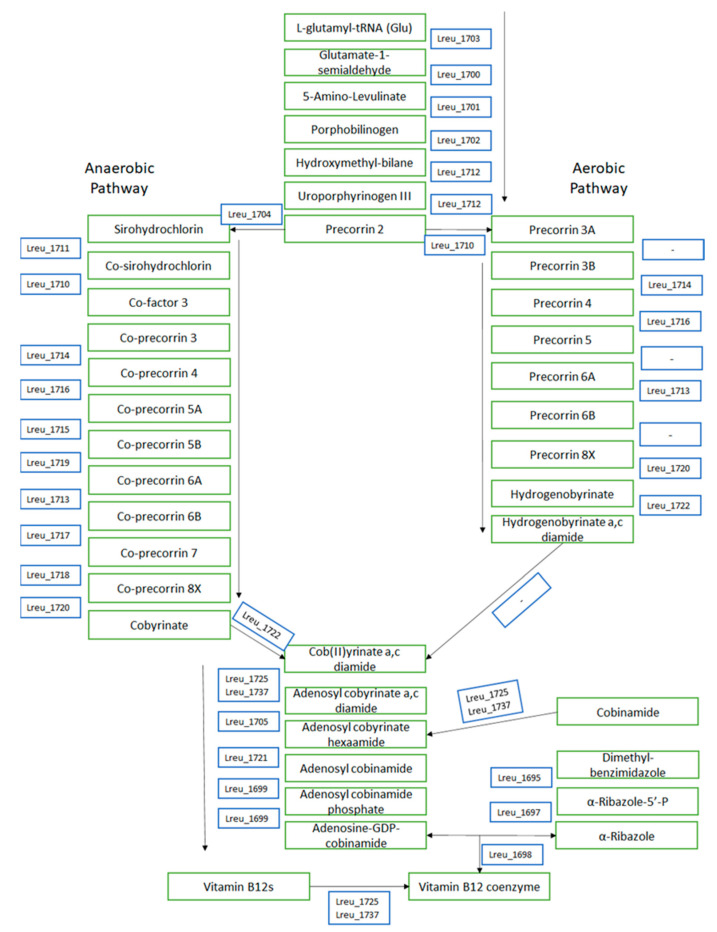
The potential pathway for cobalamin synthesis within *L. reuteri* DSM 20016, with added significance to the completed anaerobic pathway due to the bacterium being a facultative anaerobe. Compounds are listed as green, and genes are listed as blue. The KEGG pathway module for porphyrin was utilized in the creation of this diagram.

**Figure 8 microorganisms-10-01341-f008:**
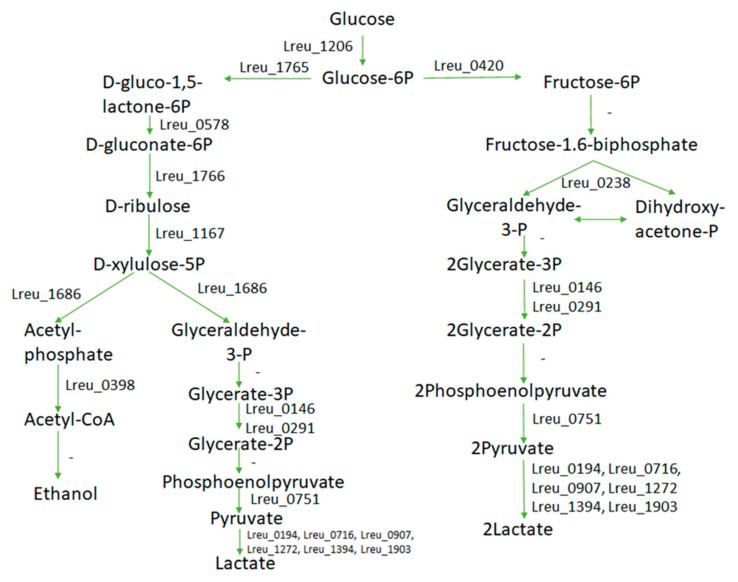
Pathway of glycolysis within *L. reuteri* DSM 20016, showcasing the EMP (right) and PKP (left) pathway and the relevant genes for each step. There are some missing genes and pathways, however. Due to the production of lactate, there is most likely an alternative mechanism.

**Figure 9 microorganisms-10-01341-f009:**
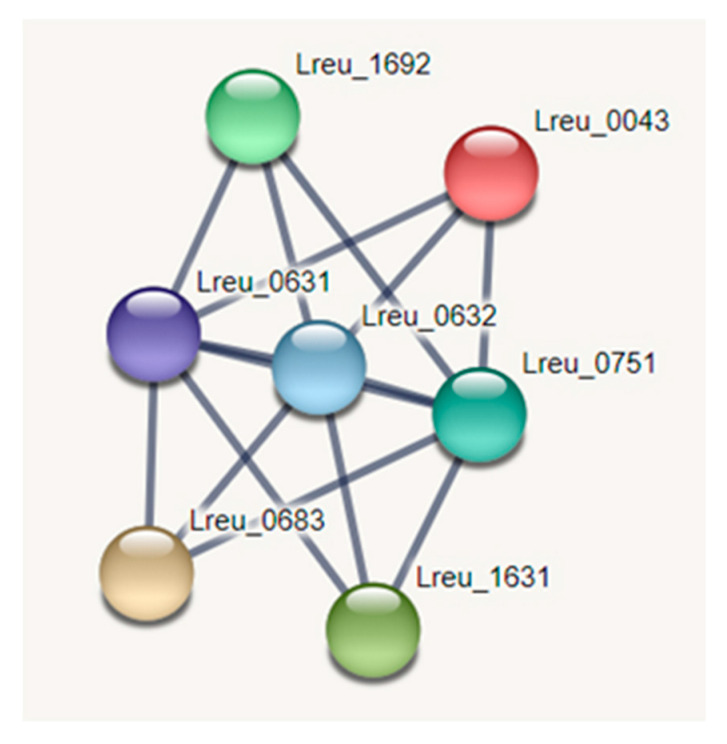
STRING network map of genes related to D-lactate dehydrogenase, involved in pyruvate synthesis (highest confidence, 0.900). Other genes showcased are that of pyruvate kinase and pyruvate dehydrogenase E1 subunit and E2 subunit proteins, which all directly relate to pyruvate, suggesting that these genes are connected. (https://string-db.org/ (accessed on 10 September 2020)) [33].

**Figure 10 microorganisms-10-01341-f010:**
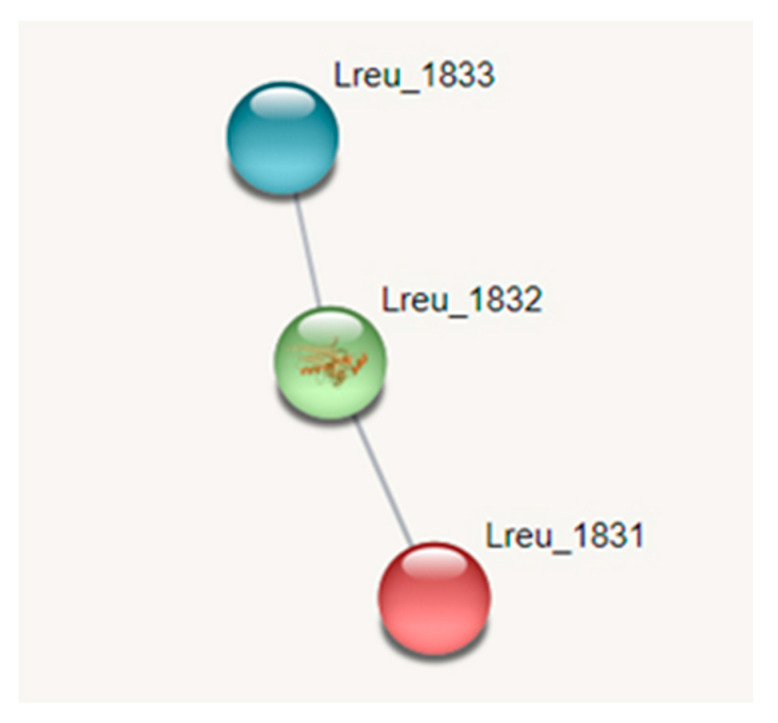
STRING network pathway map featuring the three components involved in histamine production as histidine carboxylase (medium confidence, 0.400). Low confidence despite expressed together in the hdc cluster. No regulation gene/transcriptional regulator identified. (https://string-db.org/ (accessed on 10 September 2020)) [33].

**Figure 11 microorganisms-10-01341-f011:**
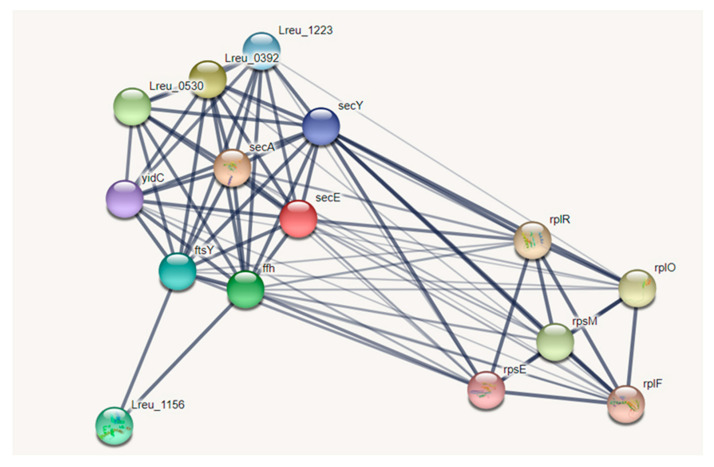
STRING network map detailing the Sec pathway (medium confidence, 0.400) along with five first shell interactors—all of which exist as ribosomal proteins. The analysis suggests that the Sec transportation pathway is closely related to translation within *L. reuteri* DSM 20016. (https://string-db.org/ (accessed on 10 September 2020)) [33].

**Figure 13 microorganisms-10-01341-f013:**
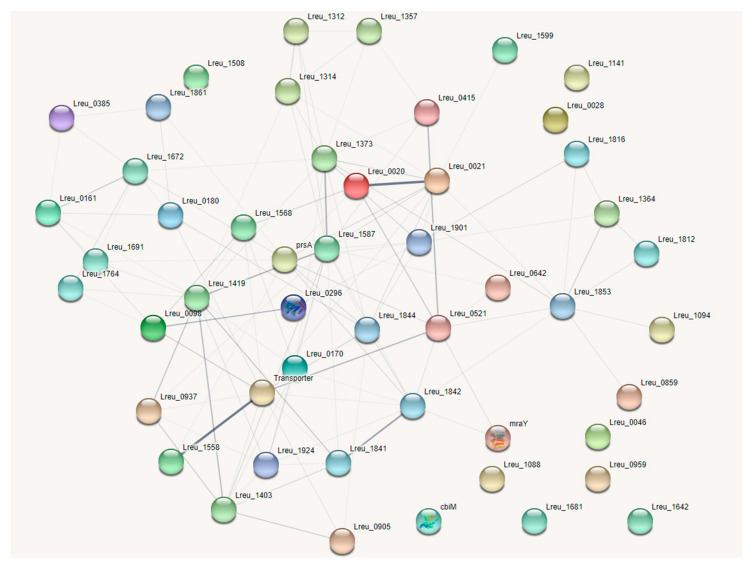
STRING network map of the proposed extracellular and transmembrane proteins within *L. reuteri* DSM 20016 (low confidence, 0.150) as per Wall et al., 2003 [81] (https://string-db.org/; (accessed on 23 April 2022) version 11.5) [82].

**Figure 14 microorganisms-10-01341-f014:**
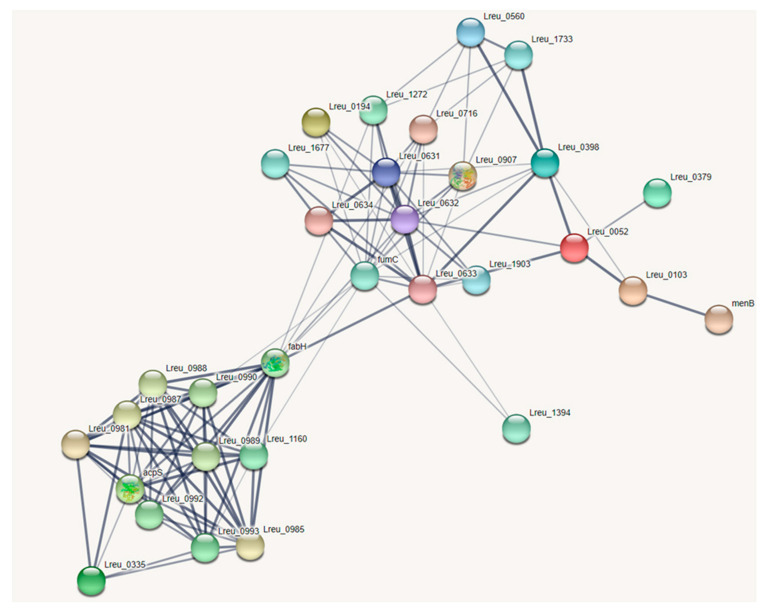
STRING network map showcasing the genes that are responsible for SCFA production within *L. reuteri* DSM 20016 (medium confidence, 0.400). There are some incomplete pathways within this strain, and no transcriptional regulators were identified. (https://string-db.org/ (accessed on 10 September 2020)) [33].

**Figure 15 microorganisms-10-01341-f015:**
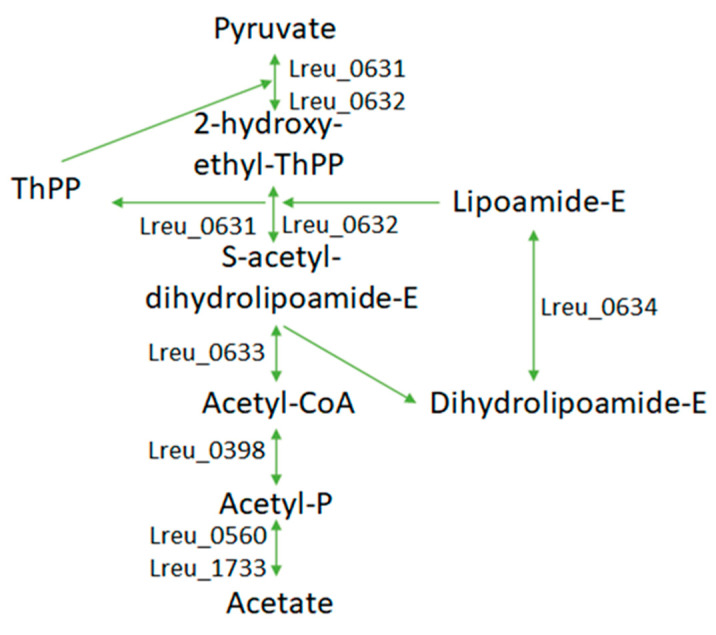
Pathway modeling the proposed pathway of acetate formation within *L. reuteri* DSM 20016, including the most likely gene candidates.

**Figure 16 microorganisms-10-01341-f016:**
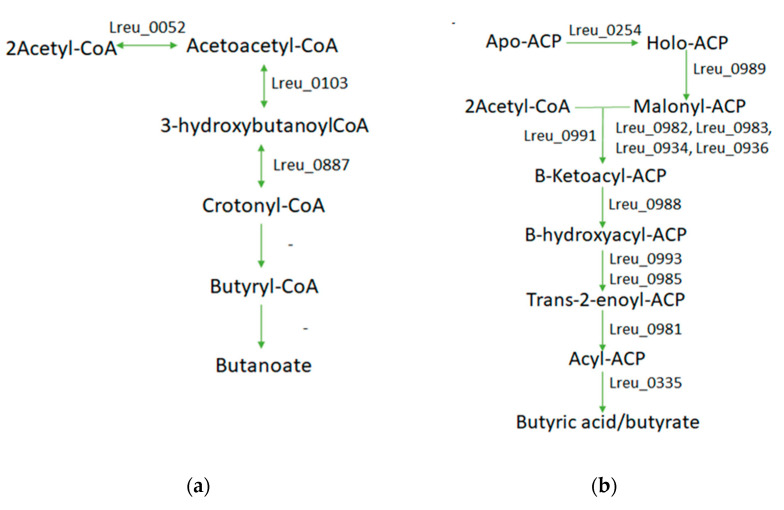
(**a**) Butanoate metabolism pathway within the bacterium, which is mostly complete. The last step may utilize a thioesterase. (**b**) Fas II cluster pathway, which can form butyryl acid and other long-chain fatty acids. Largely complete within *L. reuteri* DSM 20016.

**Figure 17 microorganisms-10-01341-f017:**
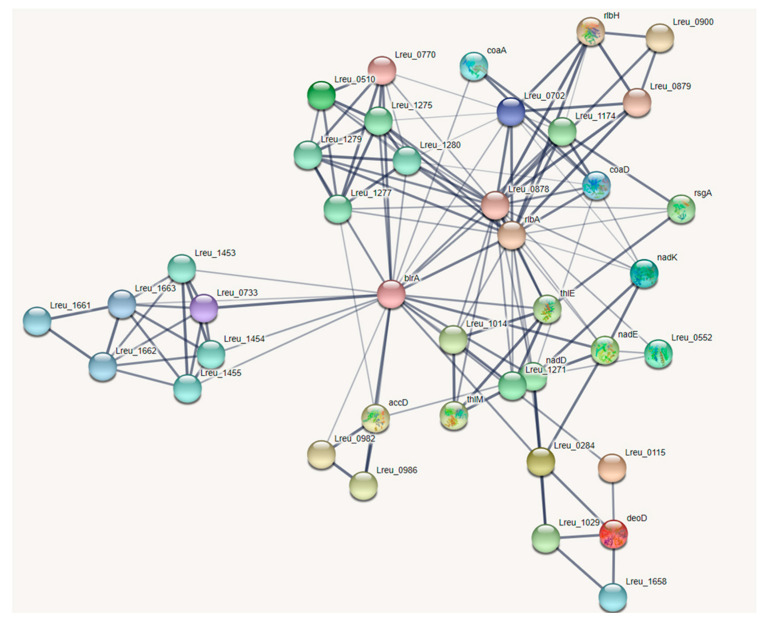
STRING network map showing the interconnected connections in the vitamin pathways of *L. reuteri* DSM 20016 (medium confidence, 0.400). Included are that of folate, thiamine, pantothenic acid, riboflavin, biotin, and niacin pathway genes. (https://string-db.org/ (accessed on 10 September 2020)) [33].

**Figure 18 microorganisms-10-01341-f018:**
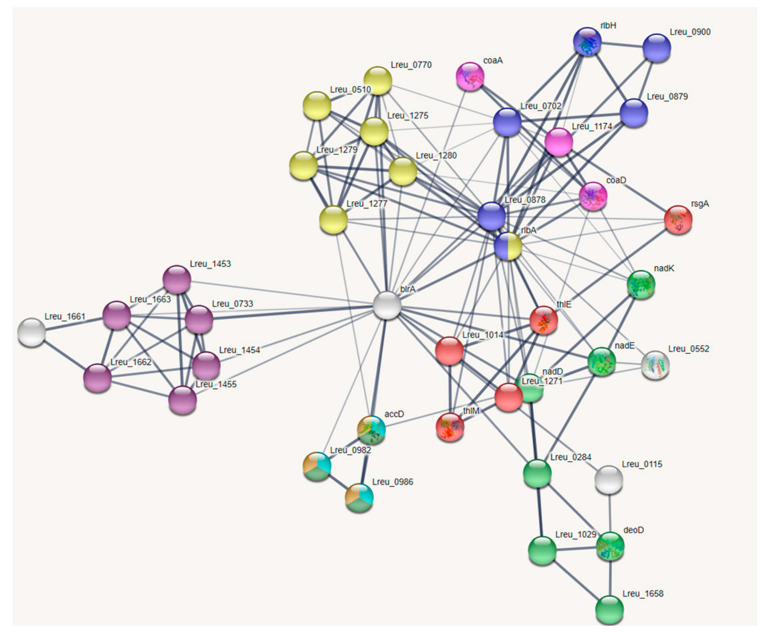
STRING network map of vitamin pathways within *L. reuteri* DSM 20016 organized by color: folate biosynthesis—yellow, nicotinate and nicotinamide metabolism—green, riboflavin metabolism—blue, thiamine metabolism—red, ABC transporters—dark purple, pantothenate and CoA biosynthesis—light purple/hot pink, fatty acid biosynthesis—light blue, propionate metabolism—dark green, and pyruvate metabolism—orange (https://string-db.org/ (accessed on 10 September 2020)) [33].

**Figure 19 microorganisms-10-01341-f019:**
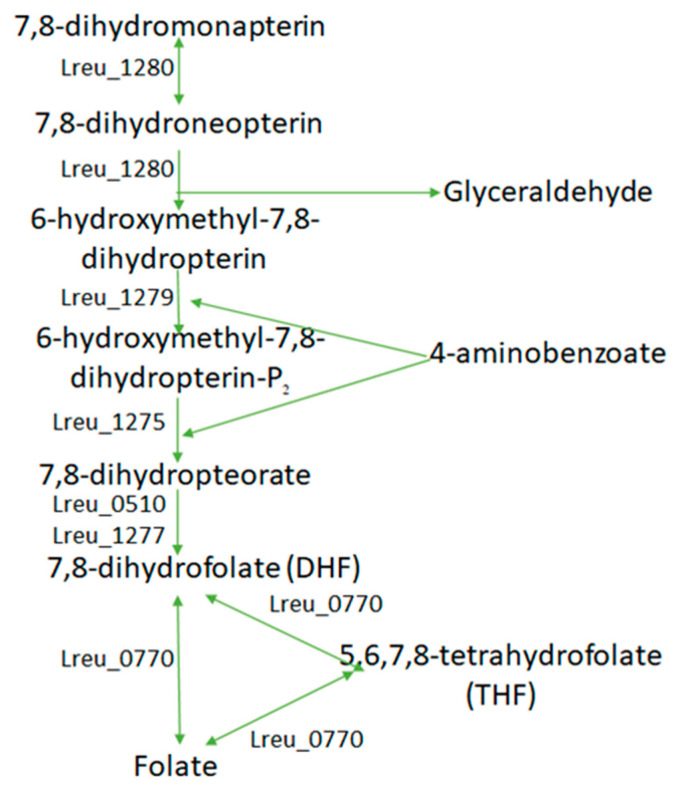
The mechanism and likely genes of which *L. reuteri* DSM 20016 may use to synthesize folate (vitamin B9). The lower part of this pathway is largely complete.

**Figure 20 microorganisms-10-01341-f020:**
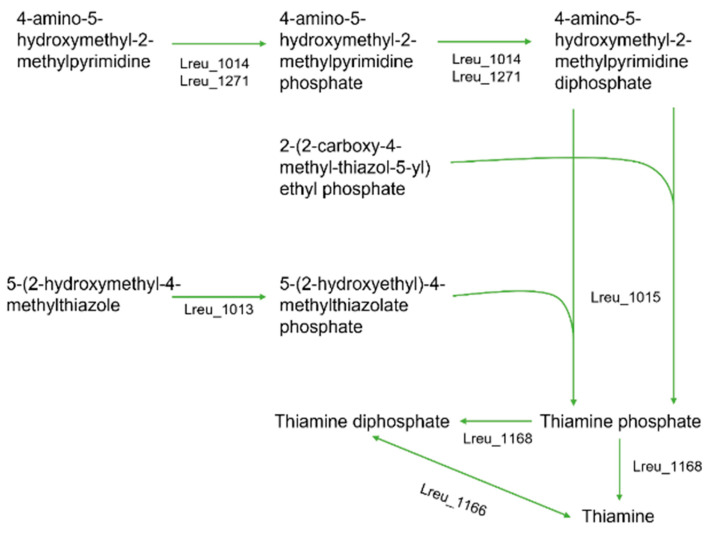
A pathway diagram that highlights the potential mechanism as to how thiamine could be synthesized within *L. reuteri* DSM 20016. The pathway does offer insight into the formation of TPP (thiamine diphosphate) from thiamine and other similar reactions.

**Figure 21 microorganisms-10-01341-f021:**
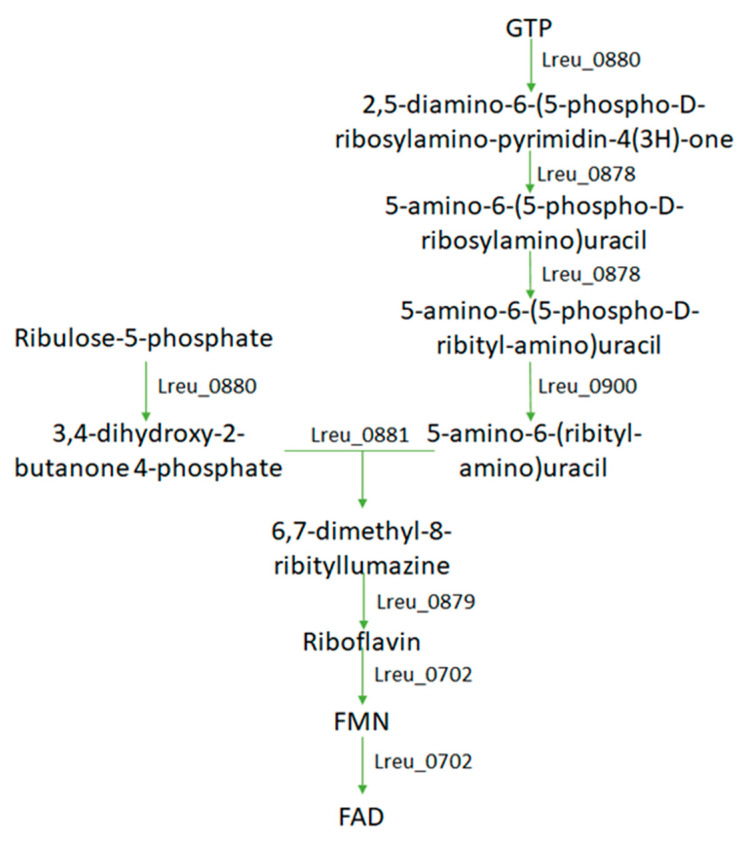
Pathway map indicating the complete synthesis of riboflavin within *L. reuteri* DSM 20016; it also shows the gene required for the formation of cofactors FMN and FAD from riboflavin.

**Figure 22 microorganisms-10-01341-f022:**
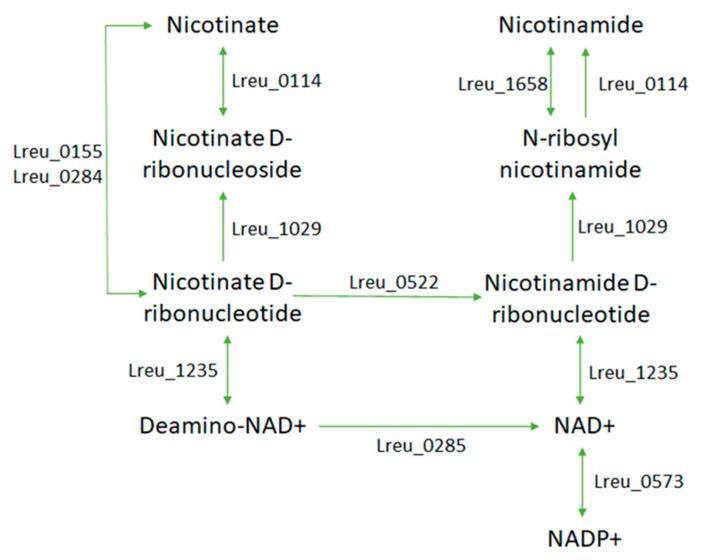
Completed pathway showing synthesis of cofactors NAD+ and NADP+ and also indicating the pathway between nicotinate and nicotinamide, suggesting the bacteria can convert between these two states.

**Figure 23 microorganisms-10-01341-f023:**
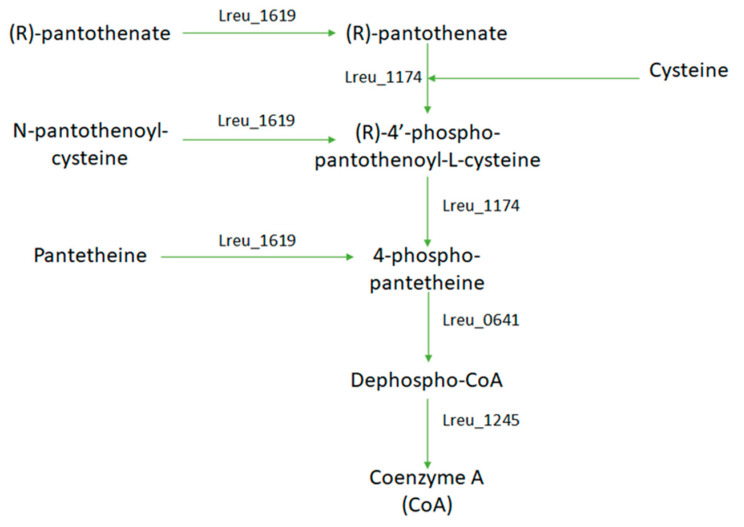
A diagram map that shows the possible formation of coenzyme A within *L. reuteri* DSM 20016 and their corresponding genes. Cysteine is required within this reaction, and this amino acid could be derived from serine.

**Figure 24 microorganisms-10-01341-f024:**
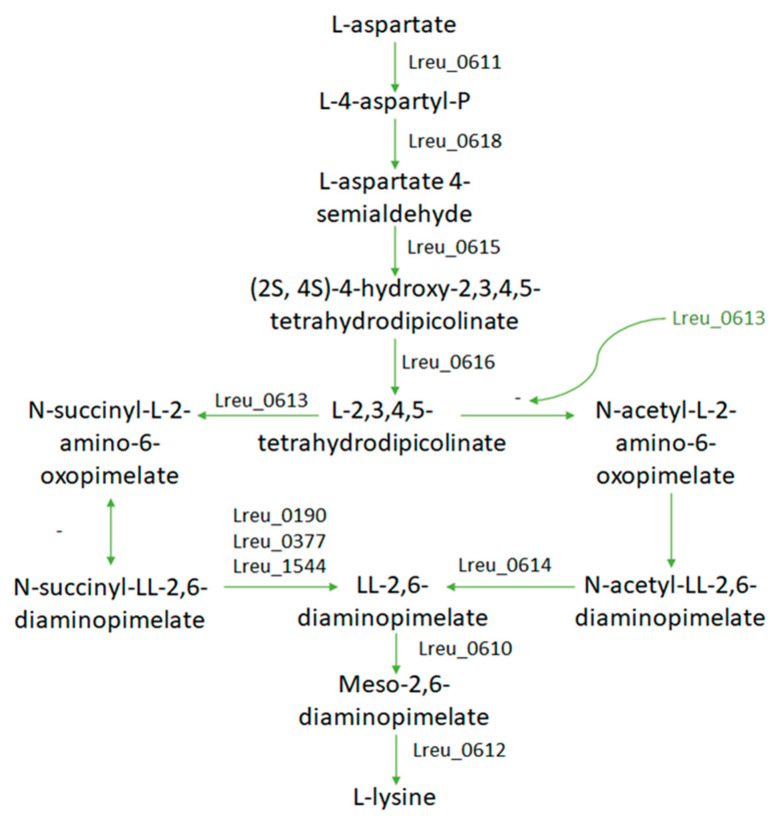
The lysine pathway module, adapted from the KEGG database. In green is the proposed gene and pathway according to KEGG and similar strain *L. reuteri* JCM 1112. With the proposed gene in place, the space between L-aspartate to lysine is completed.

**Figure 25 microorganisms-10-01341-f025:**
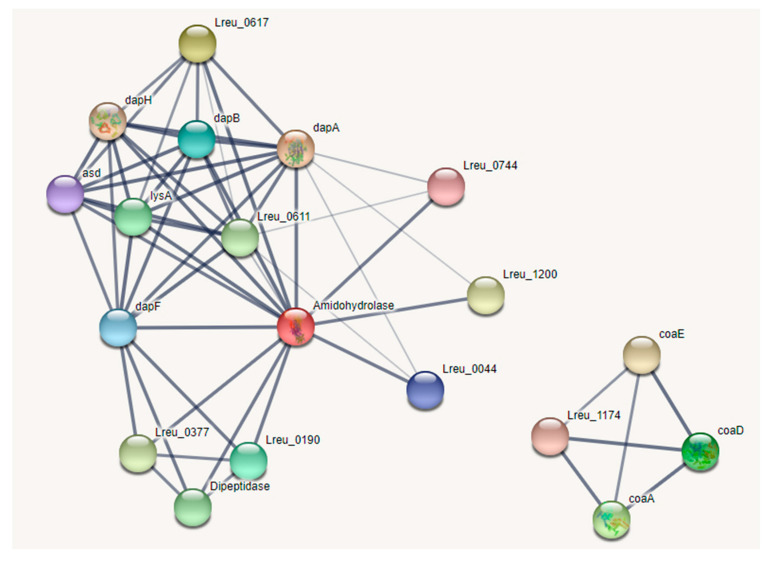
STRING network pathway of lysine biosynthesis (left) and pantothenate degradation to coenzyme A (right). Medium confidence (0.400). There are no identified transcriptional regulators within this pathway. (https://string-db.org/; (accessed on 23 April 2022) version 11.5) [82].

**Figure 26 microorganisms-10-01341-f026:**
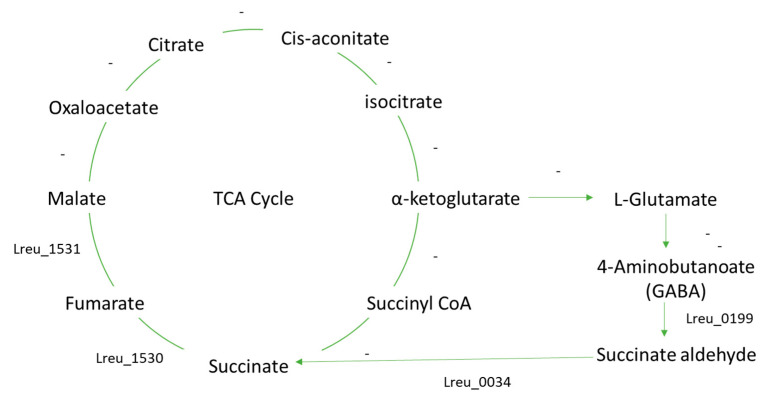
The TCA and GABA shunt pathway, as modeled from the KEGG database. The module is largely incomplete with only a few genes characterized.

**Figure 27 microorganisms-10-01341-f027:**
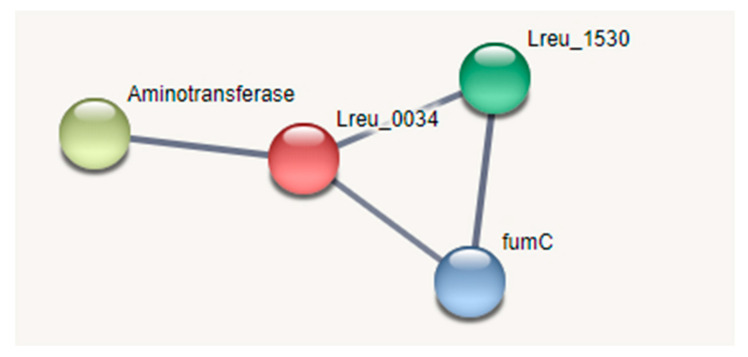
STRING network pathway of known TCA cycle and GABA shunt proteins. Medium confidence (0.400). There are no identified transcriptional regulators within this pathway. (https://string-db.org/; (accessed on 23 April 2022) version 11.5) [76].

**Table 1 microorganisms-10-01341-t001:** Genetic characteristics between *L. reuteri* DSM 20016 and *L. reuteri* JCM 1112, according to KEGG.

	*L. reuteri* DSM 20016	*L. reuteri* JCM 1112
Host	Human	Human
Chromosome size (bp)	1,999,618	2,039,414
GC content (%)	38.9	38.9
Total genes	N/A	N/A
Protein genes	1900	1820
RNA genes	88	81
Strain designation	F275	F275
Genbank	CP000705	AP007281

**Table 2 microorganisms-10-01341-t002:** Genetic characteristics between *L. reuteri* DSM 20016 and *L. reuteri* JCM 1112, according to 2021 sequenced data.

	*L. reuteri* DSM 20016	*L. reuteri* JCM 1112
Host	Human	Human
Chromosome size (bp)	1,999,618	2,039,414
GC content (%)	38.9	38.9
Total genes	2041	2069
Protein genes	1904	1912
RNA genes	89	86
Strain designation	F275	F275
NCBI Reference	NC_009513.1	NC_010609.1

**Table 3 microorganisms-10-01341-t003:** Genetic characteristics of other *L. reuteri* strains.

	SD2112	MM4-1A	KUB-AC5
Host	Human	Human	Chicken
Chromosome size (bp)	2,264,399	2,067,914	~2,190,000
Plasmids	4	0	-
GC content (%)	39.3	~38	-
Total genes	2305	2151	-
Protein genes	2082	1982	2196
RNA genes	91	116	-
Accession number	NC_015697.1	NZ_ACGX00000000.2	SRR10059212

## Data Availability

The data presented in this study are available in Appendix A.

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
