# Peer review of "In Silico Genomic and Metabolic Atlas of *Limosilactobacillus reuteri* DSM 20016: An Insight into Human Health"

_microorganisms, 2022, doi:10.3390/microorganisms10071341_

Round 1

Reviewer 1 Report

The study “In Silico Genomic and Metabolic Atlas of Limosilactobacillus reuteri DSM 20016: An Insight into Human Health” conducted genomic and proteomic bioinformatic tools to investigate the probiotic potential of a typical intestinal bacteria which provides valuable information and shows an attractive way to identify probiotic potential. In addition, the manuscript has been well written.

There are some suggestions for the authors to improve the manuscript, after that this manuscript will be recommended to publish in the journal.

1.    In the introduction, the authors mentioned mostly for the beneficial functions of probiotics, hence the article has focused to use genomic and proteomic tools to analyze the probiotic potential of the L. reuteri DSM strain, so I suggest the authors to cite some research papers based on genomic and/or proteomic views to investigate the probiotic properties.

2.    Line 40-41, 45: Write full name with parenthesis for the abbreviation of GP and HNS, IBD in the first appearance of text

3.    Line 126-131. Cite the reference from other researchers which 200 genes are most important for human host health benefits

4.    Combine the Figure 3 and Figure 4 together. Indicate the light blue color of outer and inner circles in Figure 3

5.    Line 163: Escherichia coli should be italic form, as well as Line 707 (E. coli)

6.    In Figure 6, 1,3-propanediol dehydrogenase should be changed to 1,3-PD

7.    Line 267: gram positive bacteria, herein, gram should be capital letter, Gram, as well as in Line 298

8.    Line 737. Check the site is correct: www.mdpi.com/xxx/s1

9.    Table S2 may be missed at the uploading

Author Response

Reviewer 1:

The study “In Silico Genomic and Metabolic Atlas of Limosilactobacillus reuteri DSM 20016: An Insight into Human Health” conducted genomic and proteomic bioinformatic tools to investigate the probiotic potential of a typical intestinal bacteria which provides valuable information and shows an attractive way to identify probiotic potential. In addition, the manuscript has been well written.

There are some suggestions for the authors to improve the manuscript, after that this manuscript will be recommended to publish in the journal.

  1. In the introduction, the authors mentioned mostly for the beneficial functions of probiotics, hence the article has focused to use genomic and proteomic tools to analyze the probiotic potential of the  reuteri DSM strain, so I suggest the authors to cite some research papers based on genomic and/or proteomic views to investigate the probiotic properties.

We thank you for this advice. This was an area that was not covered preliminary, and it helps to inform the audience about omics’ data and how it is being currently applied.

We have implemented 5 new references to support the use of genomic tools in the investigations of a probiotic’s potential. These references include the significance of omics’-based research, and multiple examples of how genomic analyses have been utilised in Lactobacillus and L. reuteri strains specifically. The examples include identifying stress adaptations, antimicrobial profiles and safety genes, which are key areas in determining a bacteria’s suitability as a probiotic.

Please find the corrections on lines 78-85.

  1. Line 40-41, 45: Write full name with parenthesis for the abbreviation of GP and HNS, IBD in the first appearance of text

Thank you for spotting these errors. GP, HNS and IBD have been elaborated on, and their full form can now be found within the text on lines 48, 49 and 53-54 respectively.

  1. Line 126-131. Cite the reference from other researchers which 200 genes are most important for human host health benefits

More information has been provided on why these 200 genes were selected, as it was very unclear as to why these genes are deemed the most important. The genes were selected based on categories the authors had deemed most significant in providing health benefits based on relevant literature and key areas that would allow L. reuteri DSM 20016 to persist to perform a probiotic function.

To clear up confusion, the text now discusses that the “genes were selected to aid the probiotic bacterium in its survival and/or due to its metabolites’ known effects on human health”. The text now aims to discuss which pathways were picked and aims to briefly describe why these areas were picked.

In brief, the paragraph discusses: desirable properties of probiotics; acid resistance (GABA shunt), competition/antimicrobials and adherence; the use of microcompartments, propionate metabolism and cobalamin in reuterin production; SCFAs, histamine, amino acids and vitamins; the secretory pathways to secrete these metabolites; and finally, the glycolysis and sugar metabolism pathways in the formation of SCFAs.

Corrections can be found on lines 160-174.

  1. Combine the Figure 3 and Figure 4 together. Indicate the light blue color of outer and inner circles in Figure 3

We are very appreciative of this advice. Figure 3 and 4 have been merged into a new and improved figure 3. The figure looks concise and much clearer to the audience, especially with the use of colour-coding.

Descriptions of the outer and inner blue rings have been added to the figure legend.

Please find the figure on line 179 and figure legend additions to lines 181-182.

  1. Line 163: Escherichia coli should be italic form, as well as Line 707 (E. coli)

Thank you for noticing the errors in nomenclature. The nomenclature for these bacteria names, and others, have been corrected.

The errors that you mentioned can be found corrected on lines 210 and 790.

  1. In Figure 6, 1,3-propanediol dehydrogenase should be changed to 1,3-PD

Figure 6 has now been renamed to Figure 5 due to a previous correction. The figure has been updated accordingly with your advice. Thank you.

The figure can be found on line 235.

  1. Line 267: gram positive bacteria, herein, gram should be capital letter, Gram, as well as in Line 29

Many thanks for noticing the oversight in nomenclature. There are four instances of Gram within the text. Three instances were lowercase, and these have since been capitalised.

They can be found on lines: 322, 354 and 474.

  1. Line 737. Check the site is correct: www.mdpi.com/xxx/s1

The data for the supplementary material has not been published online as of currently, so unfortunately the authors are unaware as to what the link will be until then. We thank you for your concern, however this will hopefully be updated if this manuscript is published.

  1. Table S2 may be missed at the uploading

Unfortunately, the supplementary material documents were two separate entities. It appears there were errors when uploading the manuscript originally and only supplementary material 2 was uploaded. We were contacted to upload S1, but in error only S1 made it to the submission site.  To correct this error, S1 and S2 have been placed into the same document, now labelled ‘Supplementary Material – Gene Lists’ and should be uploaded with the revised manuscript.

Thank you very much, Reviewer 1, for your feedback.

Reviewer 2 Report

In this work, the authors in silico investigated the genomic and metabolic atlas of the probiotic bacterial strain (Limosilactobacillus reuteri DSM 20016) and its role in human health. This is a highly interesting topic of research, the work is well written but it still require further minor editing before it can be accepted for publication. 

The conclusion; the last part of the abstract, should be rewritten, to become more focus about the findings of the work, rather than generalized conclusion. 

The role of probiotics in cancer incidence and prevention should be highlighted in the introduction, the authors may refer to and cite some recent works in this regards including DOI:10.15419/bmrat.v8i9.691 and doi.org/10.3390/cancers13010020.

Al though, the authors highlighted the genes that could potentially provide a health benefits to the host, the discussion should be enriched (About figure 3 and 4) with previous works and facts. 

The font size in figure 9 and 21 should adjusted, the may increase the size of the figure, so the font become consistence and readable.

Author Response

Reviewer 2:

In this work, the authors in silico investigated the genomic and metabolic atlas of the probiotic bacterial strain (Limosilactobacillus reuteri DSM 20016) and its role in human health. This is a highly interesting topic of research, the work is well written but it still require further minor editing before it can be accepted for publication.

The conclusion; the last part of the abstract, should be rewritten, to become more focus about the findings of the work, rather than generalized conclusion. 

Thank you very much for this advice! The abstract and conclusion were very much generalised, and both have been altered quite significantly to be more specific to our findings.

The abstract now highlights most of the key metabolites potentially produced by this strain (SCFAs, lactate, histamine, reuterin, folate and riboflavin). Lysine was one of the unmentioned metabolites, as the data generated by the study only offers some speculation as to its function and secretion of this amino acid cannot be proven as lysine may be utilised by the bacterium. There is also a mention of the likely adhesion genes found within the strain, noted by gene name. Following these statements is a brief sentence describing the significance of the proposed atlas. The abstract is limited to 200 words, with the word count having been met.

Please find the abstract changes on lines 13-22.

The conclusion has been completely rewritten and wrote in a much clearly and summarising manner. Attention is called to each of the pathways highlighted in the results section/Figure 3 and a comprehensive summary is given to each section of the results/discussion. In brief attention is paid to: reuterin and how this is linked to the pdu operon and propionate metabolism, including that of cobalamin synthesis and bacterial microcompartment genes; how L. reuteri DSM 20016 may adhere to human epithelial cells with the genes labelled by name, along with more speculated genes (Ef-Tu and GroEL), which need a more thorough investigation in vivo; resistance genes, which were briefly mentioned in the discussion of the safety of L. reuteri DSM 20016 and in terms of the GABA shunt; cobalamin synthesis and the potential ways glycolysis occurs in this strain; production of SCFAs in this strain and the proposed pathway for butanoate synthesis, which has only been investigated in a few other bacteria; the secretion pathway and future investigations for this; and finally the hdc operon for histamine.

From this follows a generalised paragraph discussing the future directions of this research and how the data could be utilised to improve healthcare.

Please find the amended conclusion on lines 786-834.

The role of probiotics in cancer incidence and prevention should be highlighted in the introduction, the authors may refer to and cite some recent works in this regards including DOI:10.15419/bmrat.v8i9.691 and doi.org/10.3390/cancers13010020.

Thank you for highlighting this. There was little discussion about how this bacterium could specifically aid in the treatment of cancer. Based on your advice, we have used the papers provided to first highlight how microbes can cause genome instability and contribute to multistep tumorigenesis. From here, we then discussed the use of probiotics in cancer treatment, specifically highlighting two recent works that investigate L. reuteri strains in the amelioration and prevention of colorectal cancer and melanoma tumours.

Please find the corrected material on lines 40-47.

Al though, the authors highlighted the genes that could potentially provide a health benefits to the host, the discussion should be enriched (About figure 3 and 4) with previous works and facts. 

The discussion in this area has been reworked for a previous comment (to improve clarity), but the point you have raised is very valid. 6 new references have been added to this area of the manuscript. The first reference highlights the desirable properties of probiotics; however, the subsequent references aim to support why these genes were selected. These highlight that: probiotics are known to produce bacterial metabolites that produce human host health benefits (Carding et al., 2015); that SCFAs are known to impact gut integrity, glucose homeostasis and immunomodulation (Blaak et al., 2020); that histamine in L. reuteri strains may be TNF suppressive (Thomas et al., 2012); and evidence that amino acids and vitamins that are secreted provide health benefits in the gut (Gu & Li, 2015; Yang & Liao, 2019).

The font size in figure 9 and 21 should adjusted, the may increase the size of the figure, so the font become consistence and readable.

The figures did have small font. Figure 9 (now Figure 8) was increased in size to accommodate this. Figure 21 (now Figure 20) was redrawn to have larger font and to be easier to read. The lines have also been smoothed out to give a clearer appearance to the figure.

Please find Figure 8 on line 302, and Figure 20 on line 565.

Thank you very much, Reviewer 2, for your feedback.

Reviewer 3 Report

Dear Authors,

You did nice job to develop “in silico” model of genomic and metabolic features of L reuteri strain DSM 20016. However, the manuscript could be significantly improved. From one side it is worth to consider in the manuscript genome based information for other L reuteri strains (Morita et al, Saulnier et al, Luo et al, Namrak et al). That is important for the case of strains DSM 20016 and JCM 1112. In Bergey (version 2015) and description of a new genus (Zheng et al 2020) these strains were shown as identical ones and both represented deposited versions/lines of strain F275. However, in some genome-based publications it was shown that genomes of both strain are very similar but 2 genomic islands were missing. One of the island exactly contained some of genes of sugar metabolism. This information should be shown & discussed in your manuscript. Perhaps, it is worth to make a Table about general features of genomic information (size, genes and so on) published and to make clear which one genome you used.

Minor comments

1.             Some abbreviations are not explained (GP, NHS and others)

2.             L 49 7 Fig 1 – L reuteri is not Lactobacillus anymore

3.             L 76 The whole sequence of L. reuteri DSM 20016 has been characterised – the whole sequence of genome of L …

4.             L 78 - https://www.ncbi.nlm.nih.gov/assembly/GCA_000016825.1 info shown indicated the sequence length - 1,999,618 nt.

5.             L 96 - L. reuteri DSM 20016 was searched – not L. reuteri DSM 20016 but its genome

6.             L 117 that is 2.11531 megabases (mb) in size – see comment 4

7.             L 158 – Figure 3 & Figure 4 – TCA cycle only 4 genes were identified (with GABA shunt) = basically no any enzymes found that are related to TCA cycle. That is very strange! You need to re-check this statement!

8.             L 382 Table 2S – it was missing in supplemental materials

9.             Figure 5 should be better explained in the text. DO genes of cobalamin synthesis belong to 1 cluster or are distributed?

10.         Figure 6 – should be redone. It is showing that 1,3-propanediol dehydrogenase is formed from reuterin

11.         L 188-189 – not good expression that the presence of compounds (glucose, glycerol) allow conversion or to form some metabolites

12.         Figure 7 – is it worth to cite here Sriramulu et al?

13.         L 218 – L. reuteri F275, of which DSM 20016 is a strain – not correct statement, these strains are the same but cultivation by different groups might lead to genome changes

14.         L 228 - Glucose concentration is utilised – not correct – just Glucose is utilized

15.         L 235 – this part of the manuscript could be revised after clarification which genome sequernce was used.

16.         L 388 – the part could be revised assuming that lactic acid is not SCFA.

17.         L 409 - lactaldehyde dehydrogenase – it is supposed that lactaldehyde dehydrogenase is making lactate from lactaldehyde = oxidizing aldehyde! Please, do prove that such an enzyme could operate in pyruvate reduction to lactate! Could that be malate dehydrogenase as you mentioned in Table 1S?

18.         L 623 – 625 These neurotransmitters were searched for in the L. reuteri DSM 20016 genome. There was no detection of serotonin, melatonin and acetylcholine. There was production of histamine, which has been previously discussed, and hydrogen sulphide. Hydrogen sulphide is likely formed through the formation of nitric oxide in Lactobacilli,… - How genome was searched for the neurotransmitters? What genes were responsible for sulphide formation?

Author Response

Reviewer 3:

Dear Authors,

You did nice job to develop “in silico” model of genomic and metabolic features of L reuteri strain DSM 20016. However, the manuscript could be significantly improved. From one side it is worth to consider in the manuscript genome based information for other L reuteri strains (Morita et al, Saulnier et al, Luo et al, Namrak et al). That is important for the case of strains DSM 20016 and JCM 1112. In Bergey (version 2015) and description of a new genus (Zheng et al 2020) these strains were shown as identical ones and both represented deposited versions/lines of strain F275. However, in some genome-based publications it was shown that genomes of both strain are very similar but 2 genomic islands were missing. One of the island exactly contained some of genes of sugar metabolism. This information should be shown & discussed in your manuscript. Perhaps, it is worth to make a Table about general features of genomic information (size, genes and so on) published and to make clear which one genome you used.

Thank you for comments about the genomic characterisation of L. reuteri DSM 20016. We did use this strain in our research, although unfortunately some data we received for the chromosome size was an incorrect source. This has been corrected and as per your advice, several entire sections have been added to explain these genomic differences.

The first section discusses the difference between the two strains and what the two unique regions in L. reuteri JCM 1112 encode. A table is added to show chromosome size, GC content, protein gene and RNA gene numbers, strain designation and the Genbank accension codes used.

Please find this section between lines 143- 157.

The second section discusses the absence of glycolysis genes in L. reuteri DSM 20016 due to the absence of a genomic island (unique region I). The discussion details that homologs have been found for almost all genes, except that of glyceraldehyde 3-P dehydrogenase, which is not present in the genome (this was previously confirmed in another study (Kristjansdottir et al., 2019). The study confirmed all homologs were found apart from the gene mentioned above.

Please find this section between lines 293-295.

The third section details the absence of nitrate metabolism genes in L. reuteri DSM 20016 due to its absence of unique region II. This was a sentence used to explain that the nitric oxide neurotransmitter is unlikely as it cannot be synthesised in this bacterium, at least not by any current metabolic pathway documented.

The sentence can be found on lines 693-695.

Minor comments

  1. Some abbreviations are not explained (GP, NHS and others)

Thank you. These abbreviations have been written in full-form, and care has been taken to identify further errors like this.

Please see lines 48, 49, 53-54, and 310.

  1. L 49 7 Fig 1 – L reuteri is not Lactobacillus anymore

Figure 1 has been updated accordingly, and care has been taken to ensure that all instances of L. reuteri are labelled as Limosilactobacillus where required.

Figure 1 can be found on line 87.

  1. L 76 The whole sequence of L. reuteri DSM 20016 has been characterised – the whole sequence of genome of L …

Thank you, this has been updated to read “The whole sequence of the L. reuteri DSM 20016 genome”.

Please find this on line 93.

  1. L 78 - https://www.ncbi.nlm.nih.gov/assembly/GCA_000016825.1 info shown indicated the sequence length - 1,999,618 nt.

Thank you for correcting this. The original source where this value was collected was incorrect and outdated. This was an oversight on our parts. The data has been corrected.

Please find this on line 134.

  1. L 96 - L. reuteri DSM 20016 was searched – not L. reuteri DSM 20016 but its genome

The sentence has been changed to “The genome of L. reuteri DSM 20016 was searched”. Thank you.

The change can be seen on line 108.

  1. L 117 that is 2.11531 megabases (mb) in size – see comment 4

The error has been acknowledged and this has been corrected and can be found on line 134.

  1. L 158 – Figure 3 & Figure 4 – TCA cycle only 4 genes were identified (with GABA shunt) = basically no any enzymes found that are related to TCA cycle. That is very strange! You need to re-check this statement!

The point raised is very valid. The TCA cycle on Figure 3 (which is now an amalgamation of the original figure 3+4) has been renamed as the “GABA shunt”. There has been a shift of focus onto the GABA shunt pathway and its potential acid resistance function. Research has suggested that the TCA cycle does not need to be complete and that only two genes (GABA aminotransferase and succinic semialdehyde dehydrogenase) are needed for a functional GABA shunt pathway, as seen in L. monocytogenes (Feehily et al., 2013). L. reuteri DSM 20016 features both these genes in its genome (Lreu_0199 and Lreu_0034), which would suggest that GABA is able to be metabolised and succinate synthesis can occur.

The lack of enzymes for the TCA cycle has no current explanation, but no known homologs can be found within the genome of this strain. Potentially more exploration into the proteome of L. reuteri DSM 20016 through in vivo experiments could give insights into how the TCA cycle works in this bacterium.

Please find the updated section on lines 723-734.

  1. L 382 Table 2S – it was missing in supplemental materials

There was an error with submission of the supplementary material tables. Both tables have since been put into the same document and should hopefully be submitted as part of the revisions of the manuscript. Thank you.

  1. Figure 5 should be better explained in the text. DO genes of cobalamin synthesis belong to 1 cluster or are distributed?

Thank you for your comment. The text around this figure is unclear and has been updated accordingly. The pdu operon has now been clearly discussed in this paragraph, and it is stated that the genes are encoded on this one cluster. Cobalamin synthesis genes are mentioned as “all encoded in the pdu operon”. To make the figure clearer, the text also discusses the groups of genes in the pdu operon protein connection map. The top group, which is that of propionate, cobalamin and reuterin production, and the bottom group, comprising of the bacterial microcompartment proteins

Please find these lines found on line 214-215, 217 and 218-221. The figure is now labelled as Figure 4.

  1. Figure 6 – should be redone. It is showing that 1,3-propanediol dehydrogenase is formed from reuterin

Unfortunately, this was an error, which has now been fixed. The figure (now Figure 5) and the figure legend now read as 1,3-PD (1,3-propanediol). This was an oversight with confusing the enzyme with the end product. Thank you very much for your observation.

  1. L 188-189 – not good expression that the presence of compounds (glucose, glycerol) allow conversion or to form some metabolites

These sentences were indeed very unclear. They have since been altered. They now read as “Glucose is needed in the conversion of…” and “The presence of glycerol permits…” to very clearly express that these compounds are explicitly needed in these reactions. Thank you.

Please find these on lines 239 and 240.

  1. Figure 7 – is it worth to cite here Sriramulu et al?

The reference was added due to the primary author being unsure as to whether this needed to be included, as the figure had been heavily influenced by the referenced paper. Due to the many differences between the two figures, the reference has been removed from the figure legend (now Figure 6). Many thanks.

  1. L 218 – L. reuteri F275, of which DSM 20016 is a strain – not correct statement, these strains are the same but cultivation by different groups might lead to genome changes

The sentence has been rewritten and now reads: “L. reuteri DSM 20016 is a lab culture of F275, which may have undergone genomic changes due to lab cultivation”. Our previous sentence was incorrect, so thank you very much for your correction.

Please find the described sentence on lines 268-269.

  1. L 228 - Glucose concentration is utilised – not correct – just Glucose is utilized

Thank you. The sentence has been updated to say, “Glucose is utilized…”.

The updated sentence can be found on line 279.

  1. L 235 – this part of the manuscript could be revised after clarification which genome sequernce was used.

A section has been added to discuss the significance of L. reuteri DSM 20016 missing unique region I – which is the genomic island that contains glycolysis genes in L. reuteri JCM 1112. A study (Kristjansdottir et al., 2019) suggests there are homologs for all genes apart from one, which is not present in the genome. This study could find no presence of this and two other genes, despite extensive searches. As seen in Figure 3, these appear to be distributed throughout the entire genome and are not localised.

This section can be seen on lines 292-294.

  1. L 388 – the part could be revised assuming that lactic acid is not SCFA.

The sentences have been mildly altered to mention that lactic acid is not a SCFA.

Please find these on lines 454-457.

  1. L 409 - lactaldehyde dehydrogenase – it is supposed that lactaldehyde dehydrogenase is making lactate from lactaldehyde = oxidizing aldehyde! Please, do prove that such an enzyme could operate in pyruvate reduction to lactate! Could that be malate dehydrogenase as you mentioned in Table 1S?

This was an error. Thank you so much for your observations. KEGG labels this pathway as being both a malate and L-lactate dehydrogenase. We are unsure as to how the enzyme was labelled as a lactaldehyde dehydrogenase. It is likely that this was a spelling error, rather than the database having been revised. The enzyme has now been relabelled to its proper name. The four genes that convert pyruvate to D-lactate have also been added, with the enzyme having its correct label of “D-lactate dehydrogenase”.

Please find these corrections on lines 466-468.

  1. L 623 – 625 These neurotransmitters were searched for in the L. reuteri DSM 20016 genome. There was no detection of serotonin, melatonin and acetylcholine. There was production of histamine, which has been previously discussed, and hydrogen sulphide. Hydrogen sulphide is likely formed through the formation of nitric oxide in Lactobacilli,… - How genome was searched for the neurotransmitters? What genes were responsible for sulphide formation?

The genome was searched for neurotransmitters through KEGG and through analysing a metabolic reconstruction that had been conducted on L. reuteri JCM 1112 (Kristjansdottir et al., 2019). None of the listed neurotransmitters (serotonin, melatonin or acetylcholine) were present in either.

When the manuscript was written, there was a multitude of errors made in this section. Hydrogen sulphide was confused with potential hydrogen peroxide presence, which has been further deemed to not be the case. Nitric oxide was also suggested as a potential metabolite in literature of similar strains, and as it was researched alongside hydrogen sulphide, the two statements were merged in mistake and not corrected. This was a major error, and we thank you greatly for catching this.

Neither are produced in this strain, partially because L. reuteri DSM 20016 has no nitrate metabolism genes, as these are found on unique region II in L. reuteri JCM 1112!  Instead, hydrogen sulphide is metabolised by cysteine synthase in the formation of serine, but not synthesised as the enzymes required to form it are not present in the genome. Instead, it could be said that L. reuteri is sensitive to rising hydrogen sulphide concentrations.

Please find the corrected section on lines 688-702.

Thank you very much, Reviewer 3, for your feedback.

Round 2

Reviewer 3 Report

Dear Authors,

You have significantly improved your manuscript.

1)   However, it needs to be a little bit further “polished”.

2)   I still think that you need to mention some genome – based information (for other L reuteri strains (Morita et al, Saulnier et al, Luo et al, Namrak et al).

Minor comments

L13 – The sentence - There has been some effort to characterize the metabolites

could be deleted from the abstract because this work has been done much earlier

L60 LPS – please, change to complete name

In my opinion that aim of the investigation should complete the Introduction and place right at the end of this part. I would recommend to end Introduction with the content of  LL68 – 77.

L 153 L. refuter - correct

Table 1 - it is very good to have it: please indicate/explain in the text that you found more protein genes than in JCM 1112 and what are they - unidentified ones or new ones that are not present in JCM 1112? Perhaps, it is worth to add total number of genes?

L 159 2035 genes – protein genes + ?

Figure 4 & 5 & 6.

There are some inconsistences within these Figures.

Figure 4 shows that ~ 20 genes are involved in (with) maintaining the bacterial microcompartment (bottom), where some reactions of the processes occur.

Figure 6 shows only 5 genes. What are the other doing? Structuring microcompartment or?

Chlorophyll – I think mentioning chlorophyll here (and in all manuscript) is not appropriate – Does L. reuteri produce chlorophyll?

Figure 5 shows that reuterin is produced from glycerol via PduCDE with the product – 3-HPA. Fig 6 shows that PduCDE participate in formation of propionaldehyde from 1,2-propanediol. What about reuterin?

L 239 - Glucose is needed in the conversion of 1,2-PD to propionaldehyde – it is not clear how glucose is participating in this conversion – cosubstrate? Or just a source of 1-2-PD and reducing equivalents?

General question – why PduCDE is making sometimes propionaldehyde and sometimes 3-HPA?

This part of the manuscript might be revised.

L 240 The presence of glycerol permits resting L. reuteri cells to form 3-HPA using the same nine enzymes [43]. – Not good sentence – it is not clear wjhat are nine enzymes Are these enzymes listed

L 279 Glucose is utilized by 1,2-PD degradation and metabolization of glucose to maintain a desirable NAD/NADH ratio. – not good sentence – How 1,2-PD degradation can utilize glycose? Glucose is metabolized to produce 1,2-PD and to maintain a desirable NAD/NADH ratio.

L 455 – 458. It is very convenient to present data for acids detected in mM: this shows clear what is of medium or of significant content.

Author Response

Reviewer 3

Dear Authors,

You have significantly improved your manuscript.

1)   However, it needs to be a little bit further “polished”.

2)   I still think that you need to mention some genome – based information (for other L reuteri strains (Morita et al, Saulnier et al, Luo et al, Namrak et al).

We thank you for this advice.  We were unsure as to how best to implement this. We used papers by most of the authors mentioned to highlight three key L. reuteri strains that have known human health benefits/ probiotic uses. This includes L. reuteri SD2112/ 55730 (Luo et al., 2021; Saulnier et al., 2011) , L. reuteri MM4-1A/ ATCC PTA 6475 (Saulnier et al., 2011), and L. reuteri KUB-AC5 (Nakphaichit et al., 2019; Namrak et al., 2022). We briefly discussed the use of genomic-scale models in determining probiotic potential and optimal growth conditions. The genomic information of these strains was put into Table 3.

Please find the revisions on lines 199-212.

Minor comments

L13 – The sentence - There has been some effort to characterize the metabolites

could be deleted from the abstract because this work has been done much earlier

 Thank you. The sentence has been deleted from the abstract.

L60 LPS – please, change to complete name

 LPS has been changed to Lipopolysaccharide.

The change can be found on line 59.

In my opinion that aim of the investigation should complete the Introduction and place right at the end of this part. I would recommend to end Introduction with the content of  LL68 – 77.

 The authors thank you for this comment and very much agree. The aim of the investigation (previously located on lines 68-77) has now been moved to the end of the introduction.

The end paragraph is now found on lines 80-89.

L 153 L. refuter - correct

 The error has been amended. Thank you.

Table 1 - it is very good to have it: please indicate/explain in the text that you found more protein genes than in JCM 1112 and what are they - unidentified ones or new ones that are not present in JCM 1112? Perhaps, it is worth to add total number of genes?

 The manuscript has been changed to include a secondary table that shows data from recent sequencing (2021). The recent sequencing data, found in Table 2, uses the same genomic sequences from Table 1. The text now includes two new paragraphs, with one detailing how there are variable sources for the genome sequences. The second paragraph details how the new annotations could be due to re-annotation of L. reuteri JCM 1112 using a new pipeline.

When manually analyzing the differences between the old accession number codes, many of the missing genes are uncharacterized/hypothetical. It is suggested that the genes are unidentified (due to the re-annotated sequenced data) rather than new.

Unfortunately, the KEGG database is using out-of-date data. Thankfully for L. reuteri DSM 20016, the new data only adds 4 genes and 1 pseudogene. The authors have yet to determine which new genes have been added at this time.

Total number of genes have been added to the tables.

Please find these changes from lines 176- 197.

L 159 2035 genes – protein genes + ?

 Thank you for noticing this error. This was unchanged from the previous manuscript version, where it should have been altered to match the current number of genes (protein coding) found in L. reuteri DSM 20016.

Figure 4 & 5 & 6.

There are some inconsistences within these Figures.

Figure 4 shows that ~ 20 genes are involved in (with) maintaining the bacterial microcompartment (bottom), where some reactions of the processes occur.

Figure 6 shows only 5 genes. What are the other doing? Structuring microcompartment or?

 This part of the manuscript has been altered to be clearer. The cluster points have been discussed in more detail; the bottom cluster also contains most of the propionate metabolism genes. The top cluster houses the cobalamin synthesis and cobalt genes. The bacterial microcompartment has 9 genes with the gene names amended to be included on lines 310-312.

Figure 6 shows the genes involved in propionate metabolism. The text now reflects that this is the reaction occurring in Figure 6 and that the microcompartment is the dashed hexagon. The aim was to make this section much easier to read.

Please find Figure 6 on line 316.

Chlorophyll – I think mentioning chlorophyll here (and in all manuscript) is not appropriate – Does L. reuteri produce chlorophyll?

Mentions of chlorophyll have been removed. These were included as STRING classified these genes as the porphyrin and chlorophyll biosynthesis and metabolism pathways. However, as mentioned, L. reuteri does not produce chlorophyll, so this could be confusing to some readers. Thank you very much.

Figure 5 shows that reuterin is produced from glycerol via PduCDE with the product – 3-HPA. Fig 6 shows that PduCDE participate in formation of propionaldehyde from 1,2-propanediol. What about reuterin?

 PduCDE interacts with both pathways, as it is the enzyme (+B12 cofactor) that causes both reactions. We thank you for highlighting this, as the text around this section is very confusing. We have rewritten a large portion of it. The text between Figure 5 and 6 now clearly highlights that pduCDE is utilized in both reactions and that glycerol and/or 1,2-PD are the required substrates. The products are only formed if the required substrate is present.

Both reactions likely take place in the bacterial microcompartment (due to toxicity of propionaldehyde and acrolein), suggesting pduCDE operates within the microcompartment and encounters the substrates there.

Please find the revisions on lines 301-308.

L 239 - Glucose is needed in the conversion of 1,2-PD to propionaldehyde – it is not clear how glucose is participating in this conversion – cosubstrate? Or just a source of 1-2-PD and reducing equivalents?

 Thank you for your comment. This part has been re-researched and re-written. Glucose does increase biomass metabolism, and thus provides an increase to 1,2-PD conversion. However, it’s absence or presence does not affect further conversions. The reactions occur in the resting cell state as this provides the optimum levels of glucose and other elements.

Please find amendments on lines 305-308.

General question – why PduCDE is making sometimes propionaldehyde and sometimes 3-HPA?

 We were unable to find research that suggested whether one product was made over the other. From our understanding, the products are made if their respective substrates are present, and the cell is in a resting state. So, glycerol would cause a reaction to form 3-HPA and 1,2-PD would cause a reaction to form propionaldehyde. There may be external factors that influence the rate of production, such as an optimal glucose: glycerol ratio influencing 3-HPA production.

This part of the manuscript might be revised.

 Thank you. This entire section of manuscript has been altered to give more clarity.

L 240 The presence of glycerol permits resting L. reuteri cells to form 3-HPA using the same nine enzymes [43]. – Not good sentence – it is not clear wjhat are nine enzymes Are these enzymes listed

 The enzymes were those of the bacterial microcompartment. The sentence was written poorly. Thank you very much for highlighting this. This nine-enzyme section has been moved, as noted above. The sentence “The presence of glycerol permits resting L. reuteri cells to form 3-HPA” has been rewritten to “PduCDE metabolizes glycerol to 3-HPA in resting L. reuteri cells”, which we found much clearer.

The change can be noted on line 294.

L 279 Glucose is utilized by 1,2-PD degradation and metabolization of glucose to maintain a desirable NAD/NADH ratio. – not good sentence – How 1,2-PD degradation can utilize glycose? Glucose is metabolized to produce 1,2-PD and to maintain a desirable NAD/NADH ratio.

 The sentence has been changed to say, “Glucose enhances glycerol metabolism and maintains a desirable NAD/NADH ratio, which favors an accumulation of 3-HPA”. The previous sentence inferred the wrong meaning. Glucose enhances the 1,2-PD conversion, whilst also influencing glycerol metabolism. Glucose is likely more important in the formation of SCFAs, rather than in 1,2-PD degradation, although resting L. reuteri DSM 20016 cells cultured with no glucose formed little 1,2-PD and 3-HPA (Amin et al., 2013).  3-HPA would rely on glycerol metabolism, which would be limited.

The sentence is on lines 349-350

L 455 – 458. It is very convenient to present data for acids detected in mM: this shows clear what is of medium or of significant content.

Thank you for your comment. We have added the data in mM.

The amendments can be found on lines 525-528.